# HIDDEN CONVEXITY OF WASSERSTEIN GANS: INTERPRETABLE GENERATIVE MODELS WITH CLOSED-FORM SOLUTIONS

**Arda Sahiner,**[*] **Tolga Ergen,**[*] **Batu Ozturkler, Burak Bartan, John Pauly, Morteza Mardani & Mert Pilanci**
Department of Electrical Engineering
Stanford University
Stanford, CA 94305, USA
`{sahiner,ergen,ozt,bbartan,pauly,morteza,pilanci}@stanford.edu`

## ABSTRACT

Generative Adversarial Networks (GANs) are commonly used for modeling complex distributions of data. Both the generators and discriminators of GANs are often modeled by neural networks, posing a non-transparent optimization problem which is non-convex and non-concave over the generator and discriminator, respectively. Such networks are often heuristically optimized with gradient descent-ascent (GDA), but it is unclear whether the optimization problem contains any saddle points, or whether heuristic methods can find them in practice. In this work, we analyze the training of Wasserstein GANs with two-layer neural network discriminators through the lens of convex duality, and for a variety of generators expose the conditions under which Wasserstein GANs can be solved exactly with convex optimization approaches, or can be represented as convex-concave games. Using this convex duality interpretation, we further demonstrate the impact of different activation functions of the discriminator. Our observations are verified with numerical results demonstrating the power of the convex interpretation, with applications in progressive training of convex architectures corresponding to linear generators and quadratic-activation discriminators for CelebA image generation. The code for our experiments is available at `https://github.com/ardasahiner/ProCoGAN`.

## 1 INTRODUCTION

Generative Adversarial Networks (GANs) have delivered tremendous success in learning to generate samples from high-dimensional distributions (Goodfellow et al., 2014; Cao et al., 2018; Jabbar et al., 2021). In the GAN framework, two models are trained simultaneously: a generator $G$ which attempts to generate data from the desired distribution, and a discriminator $D$ which learns to distinguish between *real* data samples and the *fake* samples generated by generator. This problem is typically posed as a zero-sum game for which the generator and discriminator compete to optimize objective $f$

$$p^* = \min_G \max_D f(G, D).$$

The ultimate goal of the GAN training problem is thus to find a saddle point (also called a Nash equilibrium) of the above optimization problem over various classes of $(G, D)$. By allowing the generator and discriminator to be represented by neural networks, great advances have been made in generative modeling and signal/image reconstruction (Isola et al., 2017; Karras et al., 2019; Radford et al., 2015; Wang et al., 2018; Yang et al., 2017). However, GANs are notoriously difficult to train, for which a variety of solutions have been proposed; see e.g., (Nowozin et al., 2016; Mescheder et al., 2018; Metz et al., 2016; Gulrajani et al., 2017).

One such approach pertains to leveraging Wasserstein GANs (WGANs) (Arjovsky et al., 2017), which utilize the Wasserstein distance with the $\ell_1$ metric to motivate a particular objective $f$. In particular, assuming that true data is drawn from distribution $p_x$, and the input to the generator is

---

[*]Equal Contribution

Table 1: Convex landscape and interpretation of WGAN with two-layer discriminator under different discriminator activation functions and generator architectures. Note that adding a linear skip connection to the discriminator imposes an additional mean matching constraint when using quadratic activation.

| Discriminator / Generator | Linear Activation | Quadratic Activation | ReLU Activation |
|---|---|---|---|
| Linear | convex | convex, closed form | convex-concave |
| 2-layer (polynomial) | convex | convex, closed form | convex-concave |
| 2-layer (ReLU) | convex | convex | convex-concave |
| Interpretation | mean matching | covariance matching | piecewise mean matching |

drawn from distribution $p_z$, we represent the generator and discriminator with parameters $\theta_g$ and $\theta_d$ respectively, to obtain the WGAN objective

$$p^* = \min_{\theta_g} \max_{\theta_d} \mathbb{E}_{\mathbf{x} \sim p_x}[D_{\theta_d}(\mathbf{x})] - \mathbb{E}_{\mathbf{z} \sim p_z}[D_{\theta_d}(G_{\theta_g}(\mathbf{z}))]. \tag{1}$$

When $G$ and $D$ are neural networks, neither the inner max, nor, the outer min problems are convex, which implies that min and max are not necessarily interchangeable. As a result, first, there is no guarantees if the saddle points exists. Second, it is unclear to what extent heuristic methods such as Gradient Descent-Ascent (GDA) for solving WGANs can approach saddle points. This lack of transparency about the loss landscape of WGANs and their convergence is of paramount importance for their utility in sensitive domains such as medical imaging. For instance, WGANs are commonly used for magnetic resonance image (MRI) reconstruction (Mardani et al., 2018; Han et al., 2018), where they can potentially hallucinate pixels and alter diagnostic decisions. Despite their prevalent utilization, GANs are not well understood.

To shed light on explaining WGANs, in this work, we analyze WGANs with two-layer neural network discriminators through the lens of convex duality and affirm that many such WGANs provably have optimal solutions which can be found with convex optimization, or can be equivalently expressed as convex-concave games, which are well studied in the literature (Žaković & Rustem, 2003; Žaković et al., 2000; Tsoukalas et al., 2009; Tsaknakis et al., 2021). We further provide interpretation into the effect of various activation functions of the discriminator on the conditions imposed on generated data, and provide convex formulations for a variety of generator-discriminator combinations (see Table 1). We further note that such shallow neural network architectures can be trained in a greedy fashion to build deeper GANs which achieve state-of-the art for image generation tasks (Karras et al., 2017). Thus, our analysis can be extended deep GANs as they are used in practice, and motivates further work into new convex optimization-based algorithms for more stable training.

**Contributions.** All in all, the main contributions of this paper are summarized as follows:

- For the first time, we show that WGAN can provably be expressed as a convex problem (or a convex-concave game) with polynomial-time complexity for two-layer discriminators and two-layer generators under various activation functions (see Table 1).
- We uncover the effects of discriminator activation on data generation through moment matching, where quadratic activation matches the covariance, while ReLU activation amounts to piecewise mean matching.
- For linear generators and quadratic discriminators, we find closed-form solutions for WGAN training as singular value thresholding, which provides interpretability.
- Our experiments demonstrate the interpretability and effectiveness of progressive convex GAN training for generation of CelebA faces.

## 1.1 RELATED WORK

The last few years have witnessed ample research in GAN optimization. While several divergence measures (Nowozin et al., 2016; Mao et al., 2017) and optimization algorithms (Miyato et al., 2018; Gulrajani et al., 2017) have been devised, GANs have not been well interpreted and the existence of saddle points is still under question. In one of the early attempts to interpret GANs, (Feizi et al., 2020) shows that for linear generators with Gaussian latent code and the 2nd order Wasserstein distance

objective, GANs coincide with PCA. Others have modified the GAN objective to implicitly enforce matching infinite-order of moments of the ground truth distribution (Li et al., 2017; Genevay et al., 2018). Further explorations have yielded specialized generators with layer-wise subspaces, which automatically discover latent "eigen-dimensions" of the data (He et al., 2021). Others have proposed explicit mean and covariance matching GAN objectives for stable training (Mroueh et al., 2017).

Regarding convergence of Wasserstein GANs, under the fairly simplistic scenario of *linear* discriminator and a two-layer ReLU-activation generator with sufficiently large width, saddle points exist and are achieved by GDA (Balaji et al., 2021). Indeed, linear discriminators are not realistic as then simply match the mean of distributions. Moreover, the over-parameterization is of high-order polynomial compared with the ambient dimension. For more realistic discriminators, (Farnia & Ozdaglar, 2020) identifies that GANs may not converge to saddle points, and for linear generators with Gaussian latent code, and continuous discriminators, certain GANs provably lack saddle points (e.g., WGANs with scalar data and Lipschitz discriminators). The findings of (Farnia & Ozdaglar, 2020) raises serious doubt about the existence of optimal solutions for GANs, though finite parameter discriminators as of neural networks are not directly addressed.

Convexity has been seldomly exploited for GANs aside from (Farnia & Tse, 2018), which studies convex duality of divergence measures, where the insights motivate regularizing the discriminator's Lipschitz constant for improved GAN performance. For supervised two-layer networks, a recent of line of work has established zero-duality gap and thus equivalent convex networks with ReLU activation that can be solved in polynomial time for global optimality (Pilanci & Ergen, 2020; Sahiner et al., 2020a; Ergen & Pilanci, 2021d; Sahiner et al., 2020b; Bartan & Pilanci, 2021; Ergen et al., 2021). These works focus on single-player networks for supervised learning. However, extending those works to the two-player GAN scenario for unsupervised learning is a significantly harder problem, and demands a unique treatment, which is the subject of this paper.

## 1.2 PRELIMINARIES

Throughout the paper, we denote matrices and vectors as uppercase and lowercase bold letters, respectively. We use $\mathbf{0}$ (or $\mathbf{1}$) to denote a vector and matrix of zeros (or ones), where the sizes are appropriately chosen depending on the context. We also use $\mathbf{I}_n$ to denote the identity matrix of size $n$. For matrices, we represent the spectral, Frobenius, and nuclear norms as $\|\cdot\|_2$, $\|\cdot\|_F$, and $\|\cdot\|_*$, respectively. Lastly, we denote the element-wise 0-1 valued indicator function and ReLU activation as $\mathbb{1}[x \geq 0]$ and $(x)_+ = \max\{x, 0\}$, respectively.

In this paper, we consider the WGAN training problem as expressed in equation 1. We consider the case of a finite real training dataset $\mathbf{X} \in \mathbb{R}^{n_r \times d_r}$ which represents the ground truth data from the distribution we would like to generate data. We also consider using finite noise $\mathbf{Z} \in \mathbb{R}^{n_f \times d_f}$ as the input to the generator as fake training inputs. The generator is given as some function $G_{\theta_g} : \mathbb{R}^{d_f} \to \mathbb{R}^{d_r}$ which maps noise from the latent space to attempt to generate realistic samples using parameters $\theta_g$, while the discriminator is given by $D_{\theta_d} : \mathbb{R}^{d_r} \to \mathbb{R}$ which assigns values depending on how realistically a particular input models the desired distribution, using parameters $\theta_d$. Then, the primary objective of the WGAN training procedure is given as

$$p^* = \min_{\theta_g} \max_{\theta_d} \mathbf{1}^\top D_{\theta_d}(\mathbf{X}) - \mathbf{1}^\top D_{\theta_d}(G_{\theta_g}(\mathbf{Z})) + \mathcal{R}_g(\theta_g) - \mathcal{R}_d(\theta_d), \tag{2}$$

where $\mathcal{R}_g$ and $\mathcal{R}_d$ are regularizers for generator and discriminator, respectively. We will analyze realizations of discriminators and generators for the saddle point problem via convex duality. One such architecture is that of the two-layer network with $m_d$ neurons and activation $\sigma$, given by

$$D_{\theta_d}(\mathbf{X}) = \sum_{j=1}^{m_d} \sigma(\mathbf{X}\mathbf{u}_j)v_j{}^1.$$

Two activation functions that we will analyze in this work include polynomial activation $\sigma(t) = at^2 + bt + c$ (of which quadratic and linear activations are special cases where $(a, b, c) = (1, 0, 0)$ and $(a, b, c) = (0, 1, 0)$ respectively), and ReLU activation $\sigma(t) = (t)_+$. As a crucial part of our convex analysis, we first need to obtain a convex representation for the ReLU activation. Therefore, we introduce the notion of hyperplane arrangements similar to (Pilanci & Ergen, 2020).

---

[1]In the case of networks with bias, one can write $D_{\theta_d}(\mathbf{X}) = \sum_{j=1}^m \sigma(\mathbf{X}\mathbf{u}_j + \mathbf{1}b_j)v_j$.

**Hyperplane arrangements**. We define the set of hyperplane arrangements as $\mathcal{H}_x := \{\operatorname{diag}(\mathbb{1}[\mathbf{X}\mathbf{u} \geq 0]) : \mathbf{u} \in \mathbb{R}^{d_r}\}$, where each diagonal matrix $\mathbf{H}_x \in \mathcal{H}_x$ encodes whether the ReLU activation is active for each data point for a particular hidden layer weight $\mathbf{u}$. Therefore, for a neuron $\mathbf{u}$, the output of the ReLU activation can be expressed as $(\mathbf{X}\mathbf{u})_+ = \mathbf{H}_x\mathbf{X}\mathbf{u}$, with the additional constraint that $(2\mathbf{H}_x - \mathbf{I}_{n_r})\mathbf{X}\mathbf{u} \geq 0$. Further, the set of hyperplane arrangements is finite, i.e. $|\mathcal{H}_x| \leq \mathcal{O}(r(n_r/r)^r)$, where $r := \operatorname{rank}(\mathbf{X}) \leq \min(n_r, d_r)$ (Stanley et al., 2004; Ojha, 2000). Thus, we can enumerate all possible hyperplane arrangements and denote them as $\mathcal{H}_x = \{\mathbf{H}_x^{(i)}\}_{i=1}^{|\mathcal{H}_x|}$. Similarly, one can consider the set of hyperplane arrangements from the generated data as $\{\mathbf{H}_g^{(i)}\}_{i=1}^{|\mathcal{H}_g|}$, or of the noise inputs to the generator: $\{\mathbf{H}_z^{(i)}\}_{i=1}^{|\mathcal{H}_z|}$. With these notions established, we now present the main results[2].

## 2 OVERVIEW OF MAIN RESULTS

As a discriminator, we consider a two-layer neural network with appropriate regularization, $m_d$ neurons, and arbitrary activation function $\sigma$. We begin with the regularized problem

$$p^* = \min_{\theta_g} \max_{v_j, \|\mathbf{u}_j\|_2 \leq 1} \sum_{j=1}^{m_d} \left[\mathbf{1}^\top \sigma(\mathbf{X}\mathbf{u}_j) - \mathbf{1}^\top \sigma(G_{\theta_g}(\mathbf{Z})\mathbf{u}_j)\right] v_j + \mathcal{R}_g(\theta_g) - \beta_d \sum_{j=1}^{m_d} |v_j| \quad (3)$$

with regularization parameter $\beta_d > 0$. This problem represents choice of $\mathcal{R}_d$ corresponding to weight-decay regularization in the case of linear or ReLU activation, and cubic regularization in the case of quadratic activation (see Appendix) (Neyshabur et al., 2014; Pilanci & Ergen, 2020; Bartan & Pilanci, 2021). Under this model, our main result is to show that with two-layer ReLU-activation generators, the solution to the WGAN problem can be reduced to convex optimization or a convex-concave game.

**Theorem 2.1.** *Consider a two-layer ReLU-activation generator of the form $G_{\theta_g}(\mathbf{Z}) = (\mathbf{Z}\mathbf{W}_1)_+\mathbf{W}_2$ with $m_g \geq n_f d_r + 1$ neurons, where $\mathbf{W}_1 \in \mathbb{R}^{d_f \times m_g}$ and $\mathbf{W}_2 \in \mathbb{R}^{m_g \times d_r}$. Then, for appropriate choice of regularizer $\mathcal{R}_g = \|G_{\theta_g}(\mathbf{Z})\|_F^2$, for any two-layer discriminator with linear or quadratic activations, the WGAN problem equation 3 is equivalent to the solution of two successive convex optimization problems, which can be solved in polynomial time in all dimensions for noise inputs $\mathbf{Z}$ of a fixed rank. Further, for a two-layer ReLU-activation discriminator, the WGAN problem is equivalent to a convex-concave game with coupled constraints.*

In practice, GANs are often solved with low-dimensional noise inputs $\mathbf{Z}$, limiting $\operatorname{rank}(\mathbf{Z})$ and enabling polynomial-time trainability. A particular example of the convex formulation of the WGAN problem in the case of a quadratic-activation discriminator can be written as

$$\mathbf{G}^* = \arg\min_{\mathbf{G}} \|\mathbf{G}\|_F^2 \text{ s.t. } \|\mathbf{X}^\top\mathbf{X} - \mathbf{G}^\top\mathbf{G}\|_2 \leq \beta_d \quad (4)$$

$$\mathbf{W}_1^*, \mathbf{W}_2^* = \arg\min_{\mathbf{W}_1, \mathbf{W}_2} \|\mathbf{W}_1\|_F^2 + \|\mathbf{W}_2\|_F^2 \text{ s.t. } \mathbf{G}^* = (\mathbf{Z}\mathbf{W}_1)_+\mathbf{W}_2, \quad (5)$$

where the solution $\mathbf{G}^*$ to equation 4 can be found in polynomial-time via singular value thresholding, formulated exactly as $\mathbf{G}^* = \mathbf{L}(\mathbf{\Sigma}^2 - \beta_d\mathbf{I})_+^{1/2}\mathbf{V}^\top$ for any orthogonal matrix $\mathbf{L}$, where $\mathbf{X} = \mathbf{U}\mathbf{\Sigma}\mathbf{V}^\top$ is the SVD of $\mathbf{X}$. While equation 5 does not appear convex, it has been shown that its solution is equivalent to a convex program (Ergen & Pilanci, 2021a; Sahiner et al., 2020a), which for the norm $\|\mathbf{S}\|_{\mathrm{K}_i,*} := \min_{t\geq 0} t \text{ s.t. } \mathbf{S} \in t\operatorname{conv}\{\mathbf{Z} = \mathbf{h}\mathbf{g}^T : (2\mathbf{H}_z^{(i)} - \mathbf{I}_{n_f})\mathbf{Z}\mathbf{u} \geq 0, \|\mathbf{Z}\|_* \leq 1\}$ is expressed as

$$\{\mathbf{V}_i^*\}_{i=1}^{|\mathcal{H}_z|} = \arg\min_{\mathbf{V}_i} \sum_{i=1}^{|\mathcal{H}_z|} \|\mathbf{V}_i\|_{\mathrm{K}_i,*} \text{ s.t. } \mathbf{G}^* = \sum_{i=1}^{|\mathcal{H}_z|} \mathbf{H}_z^{(i)}\mathbf{Z}\mathbf{V}_i, \quad (6)$$

The optimal solution to equation 6 can be found in polynomial-time in all problem dimensions when $\mathbf{Z}$ is fixed-rank, and can construct the optimal generator weights $\mathbf{W}_1^*, \mathbf{W}_2^*$ (see Appendix C.1). This WGAN problem can thus be solved in two steps: first, it solves for the optimal generator output; and second, it parameterizes the generator with ReLU weights to achieve the desired generator output. In the case of ReLU generators and ReLU discriminators, we find equivalence to a convex-concave game with coupled constraints, which we discuss further in the Appendix (Žaković & Rustem, 2003). For certain simple cases, this setting still reduces to convex optimization.

---

[2]All the proofs and some extensions are presented in Appendix.

**Theorem 2.2.** *In the case of 1-dimensional ($d_r = 1$) data $\{x_i\}_{i=1}^n$ where $n_r = n_f = n$, a two-layer ReLU-activation generator, and a two-layer ReLU-activation discriminator with bias, with arbitrary choice of convex regularizer $\mathcal{R}_g(\mathbf{w})$, the WGAN problem can be solved by first solving the following convex optimization problem*

$$\mathbf{w}^* = \arg\min_{\mathbf{w} \in \mathbb{R}^n} \mathcal{R}_g(\mathbf{w}) \text{ s.t. } \left| \sum_{i=j}^{2n} s_i(\tilde{x}_i - \tilde{x}_j) \right| \leq \beta_d, \left| \sum_{i=1}^{j} s_i(\tilde{x}_j - \tilde{x}_i) \right| \leq \beta_d, \forall j \in [2n] \quad (7)$$

*and then the parameters of the two-layer ReLU-activation generator can be found via*

$$\{(\mathbf{u}_i^*, \mathbf{v}_i^*)\}_{i=1}^{|\mathcal{H}_z|} = \arg\min_{\mathbf{u}_i, \mathbf{v}_i \in \mathcal{C}_i} \sum_{i=1}^{|\mathcal{H}_z|} \|\mathbf{u}_i\|_2 + \|\mathbf{v}_i\|_2 \text{ s.t. } \mathbf{w}^* = \sum_{i=1}^{|\mathcal{H}_z|} \mathbf{H}_z^{(i)} \mathbf{Z}(\mathbf{u}_i - \mathbf{v}_i),$$

*where*

$$\tilde{x}_i = \begin{cases} x_{\lfloor \frac{i+1}{2} \rfloor}, & \text{if } i \text{ is odd} \\ w_{\frac{i}{2}}, & \text{if } i \text{ is even} \end{cases}, \quad s_i = \begin{cases} +1, & \text{if } i \text{ is odd} \\ -1, & \text{if } i \text{ is even} \end{cases}, \quad \forall i \in [2n]$$

*for convex sets $\mathcal{C}_i$, given that the generator has $m_g \geq n+1$ neurons and $\beta_d \leq \min_{i,j \in [n]:i \neq j} |x_i - x_j|$.*

This demonstrates that even the highly non-convex and non-concave WGAN problem with ReLU-activation networks can be solved using convex optimization in polynomial time when $\mathbf{Z}$ is fixed-rank.

In the sequel, we provide further intuition about the forms of the convex optimization problems found above, and extend the results to various combinations of discriminators and generators. In the cases that the WGAN problem is equivalent to a convex problem, if the constraints of the convex problem are strictly feasible, the Slater's condition implies Lagrangian of the convex problem provably has a saddle point. We thus confirm the existence of equivalent saddle point problems for many WGANs.

## 3 TWO-LAYER DISCRIMINATOR DUALITY

Below, we provide novel interpretations into two-layer discriminator networks through convex duality.

**Lemma 3.1.** *The two-layer WGAN problem equation 3 is equivalent to the following optimization problem*

$$p^* = \min_{\theta_g} \mathcal{R}_g(\theta_g) \text{ s.t. } \max_{\|\mathbf{u}\|_2 \leq 1} |\mathbf{1}^\top \sigma(\mathbf{X}\mathbf{u}) - \mathbf{1}^\top \sigma(G_{\theta_g}(\mathbf{Z})\mathbf{u})| \leq \beta_d. \quad (8)$$

One can enumerate the implications of this result for different discriminator activation functions.

### 3.1 LINEAR-ACTIVATION DISCRIMINATORS MATCH MEANS

In the case of linear-activation discriminators, the expression in equation 8 can be greatly simplified.

**Corollary 3.1.** *The two-layer WGAN problem equation 3 with linear activation function $\sigma(t) = t$ is equivalent to the following optimization problem*

$$p^* = \min_{\theta_g} \mathcal{R}_g(\theta_g) \text{ s.t. } \|\mathbf{1}^\top \mathbf{X} - \mathbf{1}^\top G_{\theta_g}(\mathbf{Z})\|_2 \leq \beta_d. \quad (9)$$

Linear-activation discriminators seek to merely match the means of the generated data $G_{\theta_g}(\mathbf{Z})$ and the true data $\mathbf{X}$, where parameter $\beta_d$ controls how strictly the two must match. However, the exact form of the generated data depends on the parameterization of the generator and the regularization.

### 3.2 QUADRATIC-ACTIVATION DISCRIMINATORS MATCH COVARIANCES

For a quadratic-activation network, we have the following simplification.

**Corollary 3.2.** *The two-layer WGAN problem equation 3 with quadratic activation function $\sigma(t) = t^2$ is equivalent to the following optimization problem*

$$p^* = \min_{\theta_g} \mathcal{R}_g(\theta_g) \text{ s.t. } \|\mathbf{X}^\top \mathbf{X} - G_{\theta_g}(\mathbf{Z})^\top G_{\theta_g}(\mathbf{Z})\|_2 \leq \beta_d. \quad (10)$$

In this case, rather than an Euclidean norm constraint, the quadratic-activation network enforces fidelity to the ground truth distribution with a spectral norm constraint, which effectively matches the empirical covariance matrices of the generated data and the ground truth data. To combine the effect of the mean-matching of linear-activation discriminators and covariance-matching of quadratic-activation discriminators, one can consider a combination of the two.

**Corollary 3.3.** *The two-layer WGAN problem equation 3 with quadratic activation function $\sigma(t) = t^2$ with an additional unregularized linear skip connection is equivalent to the following problem*

$$p^* = \min_{\theta_g} \mathcal{R}_g(\theta_g) \text{ s.t. } \begin{array}{l} \|\mathbf{X}^\top \mathbf{X} - G_{\theta_g}(\mathbf{Z})^\top G_{\theta_g}(\mathbf{Z})\|_2 \leq \beta_d \\ \mathbf{1}^\top \mathbf{X} = \mathbf{1}^\top G_{\theta_g}(\mathbf{Z}) \end{array}. \tag{11}$$

This network thus forces the empirical means of the generated and true distribution to match exactly, while keeping the empirical covariance matrices sufficiently close. Skip connections therefore provide additional utility in WGANs, even in the two-layer discriminator setting.

### 3.3 ReLU-activation Discriminators Match Piecewise Means

In the case of the ReLU activation function, we have the following scenario.

**Corollary 3.4.** *The two-layer WGAN problem equation 3 with ReLU activation function $\sigma(t) = (t)_+$ is equivalent to the following optimization problem*

$$p^* = \min_{\theta_g} \mathcal{R}_g(\theta_g) \text{ s.t. } \max_{\substack{\|\mathbf{u}\|_2 \leq 1 \\ \left(2\mathbf{H}_x^{(j_1)} - \mathbf{I}_{n_r}\right)\mathbf{X}\mathbf{u} \geq 0 \\ \left(2\mathbf{H}_g^{(j_2)} - \mathbf{I}_{n_f}\right)G_{\theta_g}(\mathbf{Z})\mathbf{u} \geq 0}} \left| \left(\mathbf{1}^\top \mathbf{H}_x^{(j_1)} \mathbf{X} - \mathbf{1}^\top \mathbf{H}_g^{(j_2)} G_{\theta_g}(\mathbf{Z})\right)\mathbf{u} \right| \leq \beta_d, \ \forall j_1, j_2. \tag{12}$$

The interpretation of the ReLU-activation discriminator relies on the concept of hyperplane arrangements. In particular, for each possible way of separating the generated and ground truth data with a hyperplane $\mathbf{u}$ (which is encoded in the patterns specified by $\mathcal{H}_x$ and $\mathcal{H}_g$), the discriminator ensures that the means of the selected ground truth data and selected generated data are sufficiently close as determined by $\beta_d$. Thus, we can characterize the impact of the ReLU-activation discriminator as *piecewise mean matching*. Thus, unlike linear- or quadratic-activation discriminators, two-layer ReLU-activation discriminators can enforce matching of multi-modal distributions.

## 4 Generator Parameterization and Convexity

Beyond understanding the effect of various discriminators on the generated data distribution, we can also precisely characterize the WGAN objective for multiple generator architectures aside from the two-layer ReLU generators discussed in Theorem 2.1, such as for linear generators.

**Theorem 4.1.** *Consider a linear generator of the form $G_{\theta_g}(\mathbf{Z}) = \mathbf{Z}\mathbf{W}$. Then, for arbitrary choice of convex regularizer $\mathcal{R}_g(\mathbf{W})$, the WGAN problem for two-layer discriminators can be expressed as a convex optimization problem in the case of linear activation, as well as in the case of quadratic activation provided $\text{rank}(\mathbf{Z})$ is sufficiently large and $\mathcal{R}_g = \frac{\beta_g}{2}\|G_{\theta_g}(\mathbf{Z})\|_F^2$. In the case of a two-layer discriminator with ReLU activation, the WGAN problem with arbitrary choice of convex regularizer $\mathcal{R}_g(\mathbf{W})$ is equivalent to a convex-concave game with coupled constraints.*

We can then discuss specific instances of the specific problem at hand. In particular, in the case of a linear-activation discriminator, the WGAN problem with weight decay on both discriminator and generator is equivalent to the following convex program

$$p^* = \min_{\mathbf{W}} \frac{\beta_g}{2}\|\mathbf{W}\|_F^2 \text{ s.t. } \|\mathbf{1}^\top \mathbf{X} - \mathbf{1}^\top \mathbf{Z}\mathbf{W}\|_2 \leq \beta_d. \tag{13}$$

In contrast, for a quadratic-activation discriminator with regularized generator outputs,

$$p^* \geq d^* = \min_{\mathbf{G}} \frac{\beta_g}{2}\|\mathbf{G}\|_F^2 \text{ s.t. } \|\mathbf{X}^\top \mathbf{X} - \mathbf{G}^\top \mathbf{G}\|_2 \leq \beta_d, \tag{14}$$

where $\mathbf{G} = \mathbf{ZW}$, with $p^* = d^*$ under the condition that $\text{rank}(\mathbf{Z})$ is sufficiently large. In particular, allowing the SVD of $\mathbf{X} = \mathbf{U}\mathbf{\Sigma}\mathbf{V}^\top$, we define $k = \max_{k:\sigma_k^2 \geq \beta_d} k$, and note that if $\text{rank}(\mathbf{Z}) \geq k$, equality holds in (14) and a closed-form solution for the optimal generator weights exists, given by

$$\mathbf{W}^* = (\mathbf{Z}^\top \mathbf{Z})^{-\frac{1}{2}} (\mathbf{\Sigma}^2 - \beta_d \mathbf{I})_+^{\frac{1}{2}} \mathbf{V}^\top. \tag{15}$$

Lastly, for arbitrary convex regularizer $\mathcal{R}_g$, the linear generator, ReLU-activation discriminator problem can be written as the following convex-concave game

$$p^* = \min_{\mathbf{W}} \max_{\mathbf{r}_{j_1,j_2},\mathbf{r}'_{j_1 j_2}} \mathcal{R}_g(\mathbf{W}) - \beta_d \sum_{j_1,j_2} (\|\mathbf{r}_{j_1 j_2}\|_2 + \|\mathbf{r}'_{j_1 j_2}\|_2) \tag{16}$$
$$+ \sum_{j_1,j_2} \left( \mathbf{1}^\top \mathbf{H}_x^{(j_1)} \mathbf{X} - \mathbf{1}^\top \mathbf{H}_g^{(j_2)} \mathbf{ZW} \right) (\mathbf{r}_{j_1 j_2} - \mathbf{r}'_{j_1 j_2})$$
$$\text{s.t.} \quad \begin{matrix} (2\mathbf{H}_x^{(j_1)} - \mathbf{I}_n)\mathbf{X}\mathbf{r}_{j_1 j_2} \geq 0, \ (2\mathbf{H}_g^{(j_2)} - \mathbf{I}_n)\mathbf{ZW}\mathbf{r}_{j_1 j_2} \geq 0 \\ (2\mathbf{H}_x^{(j_1)} - \mathbf{I}_n)\mathbf{X}\mathbf{r}'_{j_1 j_2} \geq 0, \ (2\mathbf{H}_g^{(j_2)} - \mathbf{I}_n)\mathbf{ZW}\mathbf{r}'_{j_1 j_2} \geq 0 \end{matrix}, \quad \forall j_1 \in [|\mathcal{H}_x|], \forall j_2 \in [|\mathcal{H}_g|],$$

where we see there are bi-linear constraints which depend on both the inner maximization and the outer minimization decision variables. We now move to a more complex form of generator, which is modeled by a two-layer neural network with general polynomial activation function.

**Theorem 4.2.** *Consider a two-layer polynomial-activation generator of the form $G_{\theta_g}(\mathbf{Z}) = \sigma(\mathbf{ZW}_1)\mathbf{W}_2$ for activation function $\sigma(t) = at^2 + bt + c$ with fixed $a, b, c \in \mathbb{R}$. Define $\tilde{\mathbf{z}}_i = \begin{bmatrix} \text{vec}(\mathbf{z}_i\mathbf{z}_i^\top)^\top & b\mathbf{z}_i^\top & c \end{bmatrix}^\top$ as the lifted noise data points, in which case $G_{\theta_g}(\mathbf{Z}) = \tilde{\mathbf{Z}}\mathbf{W}$. Then, for arbitrary choice of convex regularizer $\mathcal{R}_g(\mathbf{W})$, the WGAN problem for two-layer discriminators can be expressed as a convex optimization problem in the case of linear activation, as well as in the case of quadratic activation provided $\text{rank}(\tilde{\mathbf{Z}})$ is sufficiently large and $\mathcal{R}_g = \|G_{\theta_g}(\mathbf{Z})\|_F^2$. In the case of a two-layer discriminator with ReLU activation, the WGAN problem with arbitrary choice of convex regularizer $\mathcal{R}_g(\mathbf{W})$ is equivalent to a convex-concave game with coupled constraints.*

Under the parameterization of lifted noise features, a two-layer polynomial-activation generator behaves entirely the same as a linear generator. The effect of a polynomial-activation generator is thus to provide more heavy-tailed noise as input to the generator, which provides a higher dimensional input and thus more degrees of freedom to the generator for modeling more complex data distributions.

## 5 NUMERICAL EXAMPLES

### 5.1 RELU-ACTIVATION DISCRIMINATORS

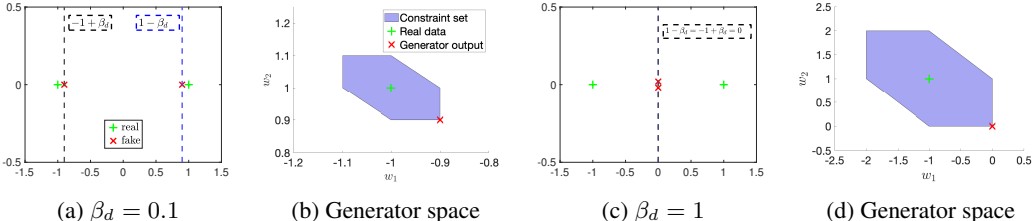

(a) $\beta_d = 0.1$      (b) Generator space      (c) $\beta_d = 1$      (d) Generator space

Figure 1: Numerical illustration of Theorem 2.2 for ReLU generator/discriminator with 1D data $\mathbf{x} = [-1, 1]^T$ and $\mathcal{R}_g(\mathbf{w}) = \|\mathbf{w}\|_2^2$. For $\beta_d = 0.1$, we observe that the constraint set of the convex program in equation 17 is a convex polyhedron shown in **(b)** and the optimal generator output is the vertex $w_1 = (-1 + \beta_d)$ and $w_2 = 1 - \beta_d$. In contrast, for $\beta_d = 1$, the constraint set in **(d)** is the larger scaled polyhedra and includes the origin. Therefore, the optimal generator output becomes $w_1 = w_2 = 0$, which corresponds to the overlapping points in **(c)** and demonstrates mode collapse.

We first verify Theorem 2.2 to elucidate the power of the convex formulation of two-layer ReLU discriminators and two-layer ReLU generators in a simple setting. Let us consider a toy dataset with

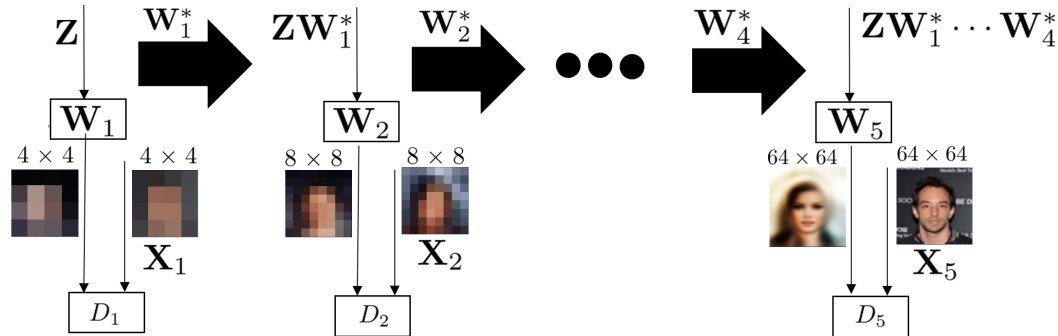

Figure 2: A modified architecture for progressive training of convex GANs (ProCoGAN). At each stage $i$, a linear generator $\mathbf{W}_i$ is used to model images at a given resolution $\mathbf{X}_i$, attempting to fool quadratic-activation discriminator $D_i$, for which the optimal solution can be found in closed-form via equation 15. Once stage $i$ is trained, the input to stage $i+1$ is given as the output of the previous stage with learned weights $\mathbf{W}_i^*$, which is then used to model higher-resolution images $\mathbf{X}_{i+1}$. The procedure continues until high-resolution images can be generated from successive application of linear generators.

the data samples $\mathbf{x} = [-1, 1]^{T3}$. Then, the convex program can be written as

$$\min_{\mathbf{w} \in \mathbb{R}^2} \mathcal{R}_g(\mathbf{w}) \ \text{s.t.} \ \left| \sum_{i=j}^{4} s_i(\tilde{x}_i - \tilde{x}_j) \right| \leq \beta_d, \ \left| \sum_{i=1}^{j} s_i(\tilde{x}_j - \tilde{x}_i) \right| \leq \beta_d, \forall j \in [4].$$

Substituting the data samples, the simplified convex problem becomes

$$\min_{\mathbf{w} \in \mathbb{R}^2} \mathcal{R}_g(\mathbf{w}) \ \text{s.t.} \ |w_1 + w_2| \leq \beta_d, \ |w_2 - 1| \leq \beta_d, \ |w_1 + 1| \leq \beta_d. \tag{17}$$

As long as $\mathcal{R}_g(\mathbf{w})$ is convex in $\mathbf{w}$, this is a convex optimization problem. We can numerically solve this problem with various convex regularization functions, such as $\mathcal{R}_g(\mathbf{w}) = \|\mathbf{w}\|_p^p$ for $p \geq 1$.

We visualize the results in Figure 1. Here, we observe that when $\beta_d = 0.1$, the constraint set is a convex polyhedron and the optimal generator outputs are at the boundary of the constraint set, i.e., $w_1 = (-1 + \beta_d)$ and $w_2 = 1 - \beta_d$. However, selecting $\beta_d = 1$ enlarges the constraint set such that the origin becomes a feasible point. Thus, due to having $\mathcal{R}_g(\mathbf{w}) = \|\mathbf{w}\|_2^2$ in the objective, both outputs get the same value $w_1 = w_2 = 0$, which demonstrates the mode collapse issue.

## 5.2 PROGRESSIVE TRAINING OF LINEAR GENERATORS AND QUADRATIC DISCRIMINATORS

Here, we demonstrate a proof-of-concept example for the simple covariance-matching performed by a quadratic-activation discriminator for modeling complex data distributions. In particular, we consider the task of generating images from the CelebFaces Attributes Dataset (CelebA) (Liu et al., 2015), using *only a linear generator and quadratic-activation discriminator*. We compare the generated faces from our convex closed-form solution in equation 15 with the ones generated using the original non-convex and non-concave formulation. GDA is used for solving the non-convex problem.

We proceed by progressively training the generators layers. This is typically used for training GANs for high-resolution image generation (Karras et al., 2017). The training operates in stages of successively increasing the resolution. In the first stage, we start with the Gaussian latent code $\mathbf{Z} \in \mathbb{R}^{n_f \times d_f}$ and locally match the generator weight $\mathbf{W}_1$ to produce samples from downsampled distribution of images $\mathbf{X}_1$. The second stage then starts with latent code $\mathbf{Z}_2$, which is the upsampled version of the network output from the previous stage $\mathbf{ZW}_1^*$. The generator weight $\mathbf{W}_2$ is then trained to match higher resolution $\mathbf{X}_2$. The procedure repeats until full-resolution images are obtained. Our approach is illustrated in Figure 2. The optimal solution for each stage can be found in closed-form using equation 15; we compare using this closed-form solution, which we call Progressive Convex GAN (ProCoGAN), to training the non-convex counterpart with Progressive GDA.

---

[3]See Appendix for derivation, where we also provide an example with the data samples $\mathbf{x} = [-1, 0, 1]^T$.

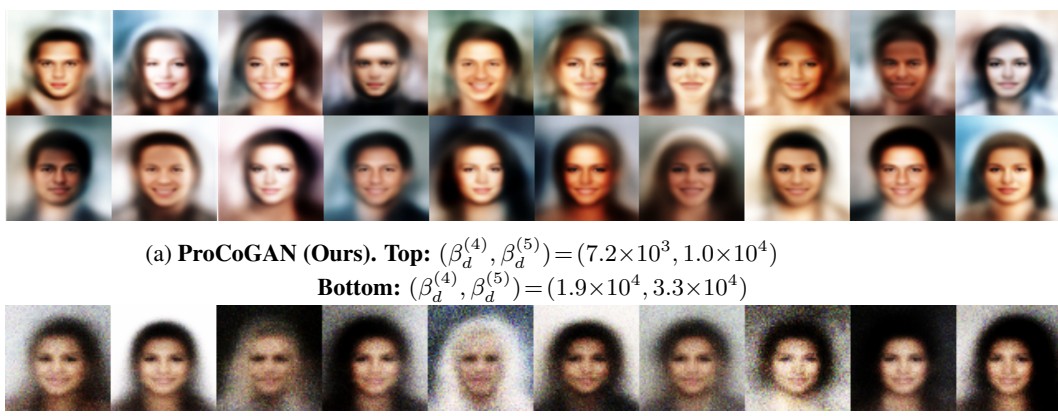

(a) **ProCoGAN (Ours). Top:** $(\beta_d^{(4)}, \beta_d^{(5)}) = (7.2 \times 10^3, 1.0 \times 10^4)$
**Bottom:** $(\beta_d^{(4)}, \beta_d^{(5)}) = (1.9 \times 10^4, 3.3 \times 10^4)$

(b) **Progressive GDA (Baseline)**

Figure 3: Representative generated faces from ProCoGAN and Progressive GDA with stagewise training of *linear* generators and quadratic-activation discriminators on CelebA (Figure 2). ProCoGAN only employs the closed-form expression equation 15, where $\beta_d$ controls the variation and smoothness in the generated images.

In practice, the first stage begins with $4 \times 4$ resolution RGB images, i.e. $\mathbf{X}_1 \in \mathbb{R}^{n_r \times 48}$, and at each successive stage we increase the resolution by a factor of two, until obtaining the final stage of $64 \times 64$ resolution. For ProCoGAN, at each stage $i$, we use a fixed penalty $\beta_d^{(i)}$ for the discriminator, while GDA is trained with a standard Gradient Penalty (Gulrajani et al., 2017). At each stage, GDA is trained with a sufficiently wide network with $m_d^{(i)} = (192, 192, 768, 3092, 3092)$ neurons at each stage, with fixed minibatches of size 16 for 15000 iterations per stage. As a final post-processing step to visualize images, because the linear generator does not explicitly enforce pixel values to be feasible, for both ProCoGAN and the baseline, we apply histogram matching between the generated images and the ground truth dataset (Shen, 2007). For both ProCoGAN and the baseline trained on GPU, we evaluate the wall-clock time for three runs. **While ProCoGAN trains for only $153 \pm 3$ seconds, the baseline using Progressive GDA takes $11696 \pm 81$ seconds to train.** ProCoGAN is much faster than the baseline, which demonstrates the power of the equivalent convex formulation.

We also visualize representative freshly generated samples from the generators learned by both approaches in Figure 3. We keep $(\beta_d^{(1)}, \beta_d^{(2)}, \beta_d^{(3)})$ fixed, and visualize the result of training two different sets of values of $(\beta_d^{(4)}, \beta_d^{(5)})$ for ProCoGAN. We observe that ProCoGAN can generate reasonably realistic looking and diverse images. The trade off between diversity and image quality can be tweaked with the regularization parameter $\beta_d$. Larger $\beta_d$ generate images with higher fidelity but with less degree of diversity, and vice versa (see more examples in Appendix B.2). Note that we are using a simple linear generator, which by no means compete with state-of-the-art deep face generation models. The interpretation of singular value thresholding per generator layer however is insightful to control the features playing role in face generation. Further evidence and more quantitative evaluation is provided in Appendix B.2. We note that the progressive closed-form approach of ProCoGAN may also provide benefits in initializing deep non-convex GAN architectures for improved convergence speed, which has precedence in the greedy layerwise learning literature (Bengio et al., 2007).

## 6 CONCLUSIONS

We studied WGAN training problem under the setting of a two-layer neural network discriminator, and found that for a variety of activation functions and generator parameterizations, the solution can be found via either a convex program or as the solution to a convex-concave game. Our findings indicate that the discriminator activation directly impacts the generator objective, whether it be mean matching, covariance matching, or piecewise mean matching. Furthermore, for the more complicated setting of ReLU activation in both two-layer generators and discriminators, we establish convex equivalents for one-dimensional data. To the best of our knowledge, this is the first work providing theoretically solid convex interpretations for non-trivial WGAN training problems, and even achieving closed-form solutions in certain relevant cases. In the light of our results and existing convex duality analysis for deeper networks, e.g., Ergen & Pilanci (2021b;c); Wang et al. (2021), we conjecture that a similar analysis can also be applied to deeper networks and other GANs.

## 7 ACKNOWLEDGEMENTS

This work was partially supported by the National Science Foundation under grants ECCS-2037304, DMS-2134248, the Army Research Office, and the National Institutes of Health under grants R01EB009690 and U01EB029427.

## 8 ETHICS AND REPRODUCIBILITY STATEMENTS

This paper aims to provide a complete theoretical characterization for the training of Wasserstein GANs using convex duality. Therefore, we believe that there aren't any ethical concerns regarding our paper. For the sake of reproducibility, we provide all the experimental details (including preprocessing, hyperparameter optimization, extensive ablation studies, hardware requirements, and all other implementation details) in Appendix B as well as the source (`https://github.com/ardasahiner/ProCoGAN`) to reproduce the experiments in the paper. Similarly, all the proofs and explanations regarding our theoretical analysis and additional supplemental analyses can be found in Appendices C, D, E, and F.

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

# Appendix

## Table of Contents

## A    NOTATION SUMMARY

Here, we provide a table summarizing the notation used in this paper for clarity.

Table 2: Notation used throughout the paper.

| Symbol | Meaning |
|---|---|
| $\theta_g$ | Generator parameters |
| $\theta_d$ | Discriminator parameters |
| $G_{\theta_g}$ | Generator function |
| $D_{\theta_d}$ | Discriminator function |
| $\mathbf{X}$ | Ground truth data |
| $\mathbf{Z}$ | Noise inputs to generator |
| $\mathcal{R}_g$ | Generator regularization function |
| $\mathcal{R}_d$ | Discriminator regularization function |
| $\beta_g$ | Generator regularization parameter |
| $\beta_d$ | Discriminator regularization parameter |
| $n_r$ | Number of samples from $\mathbf{X}$ |
| $d_r$ | Dimension of samples from $\mathbf{X}$ |
| $n_f$ | Number of samples from $\mathbf{Z}$ |
| $d_f$ | Dimension of samples from $\mathbf{Z}$ |
| $\mathcal{H}_x$ | Hyperplane arrangements of $\mathbf{X}$ |
| $\mathcal{H}_z$ | Hyperplane arrangements of $\mathbf{Z}$ |
| $r$ | rank($\mathbf{X}$) |
| $r_z$ | rank($\mathbf{Z}$) |
| $m_g$ | Generator hidden layer neurons |
| $m_d$ | Discriminator hidden layer neurons |
| $\mathbf{u}_j$ | First-layer weight of discriminator neuron $j$ |
| $v_j$ | Second-layer weight of discriminator neuron $j$ |
| $\mathbf{W}_1$ | First-layer generator weight matrix |
| $\mathbf{W}_2$ | Second-layer generator weight matrix |

## B    EXPERIMENTAL DETAILS AND ADDITIONAL NUMERICAL EXAMPLES

### B.1    ReLU-ACTIVATION DISCRIMINATORS

We first provide some non-convex experimental results to support our claims in Theorem 2.2. For this case, we use a WGAN with two-layer ReLU network generator and discriminator with the parameters $(m_g, m_d, \beta_d, \mu) = (150, 150, 10^{-3}, 4e - 6)$. We then train this architecture on the same dataset in Figure 1. As illustrated in Figure 4, depending on the initialization seed, the training performance for the non-convex architecture might significantly change. However, whenever the non-convex approach achieves a stable training performance its results match with our theoretical predictions in Theorem 2.2.

In order to illustrate how the constraints in Theorem 2.2 change depending on the number of data samples, below, we analyze a case with three data samples.

Let us consider a toy dataset with the data samples $\mathbf{x} = [-1, 0, 1]^T$. Then, the convex program can be written as

$$\min_{\mathbf{w} \in \mathbb{R}^3} \mathcal{R}_g(\mathbf{w}) \text{ s.t. } \left| \sum_{i=j}^{6} s_i(\tilde{x}_i - \tilde{x}_j) \right| \leq \beta_d, \left| \sum_{i=1}^{j} s_i(\tilde{x}_j - \tilde{x}_i) \right| \leq \beta_d, \forall j \in [6]. \tag{18}$$

Substituting the data samples, the simplified convex problem admits

$$\min_{\mathbf{w} \in \mathbb{R}^3} \mathcal{R}_g(\mathbf{w}) \text{ s.t. } \begin{aligned} &|w_1 + w_2 + w_3| \leq \beta_d, \\ &|1 - (w_2 + w_3)| \leq \beta_d, |w_1 + w_2 + 1| \leq \beta_d, \\ &|w_3 - 1| \leq \beta_d, |w_1 + 1| \leq \beta_d \end{aligned} \tag{19}$$

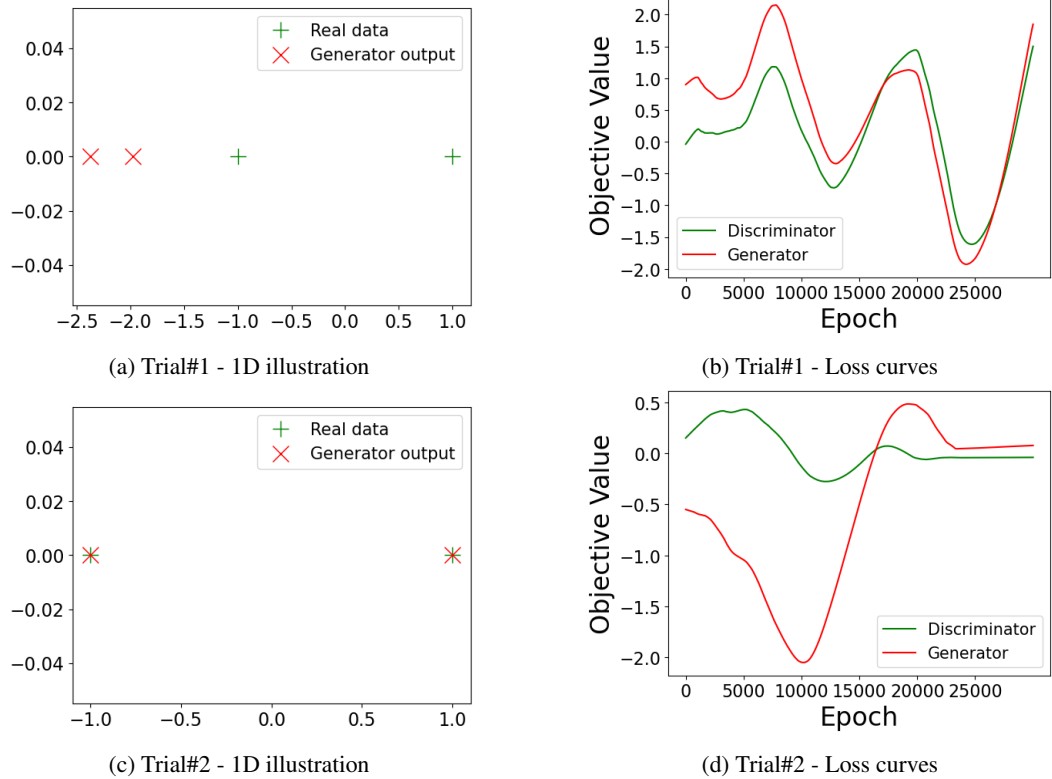

(a) Trial#1 - 1D illustration

(b) Trial#1 - Loss curves

(c) Trial#2 - 1D illustration

(d) Trial#2 - Loss curves

Figure 4: Non-convex architecture trained on the dataset in Figure 1 using the Adam optimizer with $(m_g, m_d, \beta_d, \mu) = (150, 150, 10^{-3}, 4e - 6)$. Unlike our stable convex approach, the non-convex training is unstable and leads to undamped oscillations depending on the initialization. In particular, for Trial#1 ((**a**) and (**b**)), we obtain unstable training so that the generator is unable to capture the trend in the real data. However, in Trial#2 ((**c**) and (**d**)), the non-convex architecture is able to match the real data as predicted our theory in Theorem 2.2.

which exhibits similar trends (compared to the case with two samples in Figure 1) as illustrated in Figure 5.

*Proof.* To derive the convex form, we begin with equation 18 and simplify to:

$j = 1 \quad |-(w_1 + 1) + 1 - (w_2 + 1) + 2 - (w_3 + 1)| \le \beta_d \quad 0 \le \beta_d$

$j = 2 \quad |-w_1 - (w_2 - w_1) + (1 - w_1) - (w_3 - w_1)| \le \beta_d \quad |w_1 + 1| \le \beta_d$

$j = 3 \quad |-w_2 + 1 - w_3| \le \beta_d \quad |1 + w_1| \le \beta_d$

$j = 4 \quad |(1 - w_2) - (w_3 - w_2)| \le \beta_d \quad |w_2 - (w_2 - w_1) + (w_2 + 1)| \le \beta_d$

$j = 5 \quad |w_3 - 1| \le \beta_d \quad |2 - (1 - w_1) + 1 - (1 - w_2)| \le \beta_d$

$j = 6 \quad 0 \le \beta_d \quad |(w_3 + 1) - (w_3 - w_1) + w_3$
$\qquad\qquad\qquad\qquad - (w_3 - w_2) + (w_3 - 1)| \le \beta_d.$

Simplifying the constraints above yield

$\begin{aligned} j = 1 \quad &|w_1 + w_2 + w_3| \le \beta_d & 0 \le \beta_d \\ j = 2 \quad &|1 - (w_2 + w_3)| \le \beta_d & |w_1 + 1| \le \beta_d \\ j = 3 \quad &|1 - (w_2 + w_3)| \le \beta_d & |w_1 + 1| \le \beta_d \\ j = 4 \quad &|w_3 - 1| \le \beta_d & |w_1 + w_2 + 1| \le \beta_d \\ j = 5 \quad &|w_3 - 1| \le \beta_d & |w_1 + w_2 + 1| \le \beta_d \\ j = 6 \quad &0 \le \beta_d & |w_1 + w_2 + w_3| \le \beta_d. \end{aligned}$

which can further be simplified to the expression in equation 19. $\qquad\square$

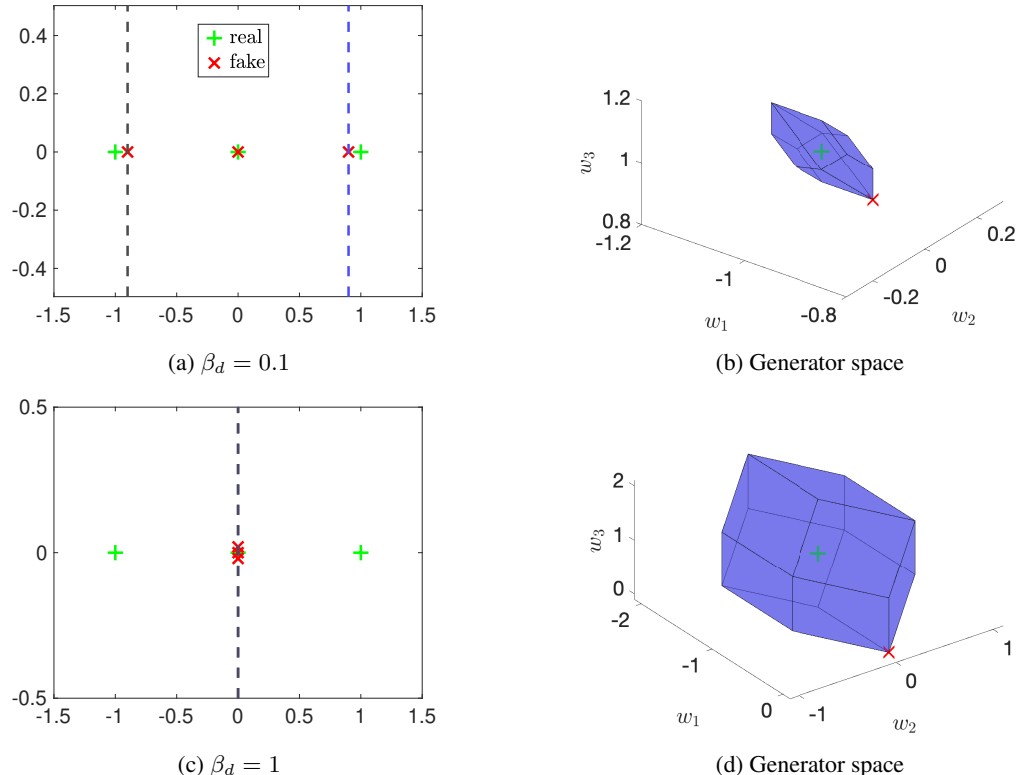

Figure 5: Numerical illustration of Theorem 2.2 for ReLU generator/discriminator with 1D data $\mathbf{x} = [-1, 0, 1]^T$ and $\mathcal{R}_g(\mathbf{w}) = \|\mathbf{w}\|_2^2$.

## B.2 PROGRESSIVE TRAINING OF LINEAR GENERATORS AND QUADRATIC DISCRIMINATORS

The CelebA dataset is large-scale face attributes dataset with 202599 RGB images of resolution $218 \times 178$, which is allowed for non-commercial research purposes only. For this work, we take the first 50000 images from this dataset, and re-scale images to be square at size $64 \times 64$ as the high-resolution baseline $\mathbf{X}_5 \in \mathbb{R}^{50000 \times 12288}$. All images are represented in the range $[0, 1]$. In order to generate more realistic looking images, we subtract the mean from the ground truth samples prior to training and re-add it in visualization. The inputs to the generator network $\mathbf{Z} \in \mathbb{R}^{50000 \times 48}$ are sampled from i.i.d. standard Gaussian distribution.

For the Progressive GDA baseline, we train the networks using Adam (Kingma & Ba, 2014), with $\alpha = 1e - 3$, $\beta_1 = 0$, $\beta_2 = 0.99$ and $\epsilon = 10^{-8}$, as is done in (Karras et al., 2017). Also following (Karras et al., 2017), we use WGAN-GP loss with parameter $\lambda = 10$ and an additional penalty $\epsilon_{\text{drift}}\mathbb{E}_{x \sim p_x}[D(x)^2]$, where $\epsilon_{\text{drift}} = 10^{-3}$. Also following (Karras et al., 2017), for visualizing the generator output, we use an exponential running average for the weights of the generator with decay 0.999. For progressive GDA, similar to the ProCoGAN formulation, we penalize the outputs of the generator $G$ with penalty $\beta_g\|G\|_F^2$ for some regularization parameter $\beta_g$. For the results in the main paper, we let $\beta_g^{(i)} = 100/d_r^{(i)}$ where $d_r^{(i)}$ is the dimension of the real data at each stage $i$. At each stage of the progressive process, the weights of the previous stages are held constant and not fine-tuned, so as to match the architecture of ProCo-GAN. We plot the loss curves of the final stage of the baseline in Figure 6 to demonstrate convergence.

We emphasize that the results of Progressive GDA as shown in this paper are not identical to the original progressive training formulation of (Karras et al., 2017), with many key differences which prevent our particular architecture from generating state-of-the-art images on par with (Karras et al., 2017). Many key aspects of (Karras et al., 2017) are not captured by the architecture studied in this work, including: using higher-resolution ground truth images (up to $1024 \times 1024$),

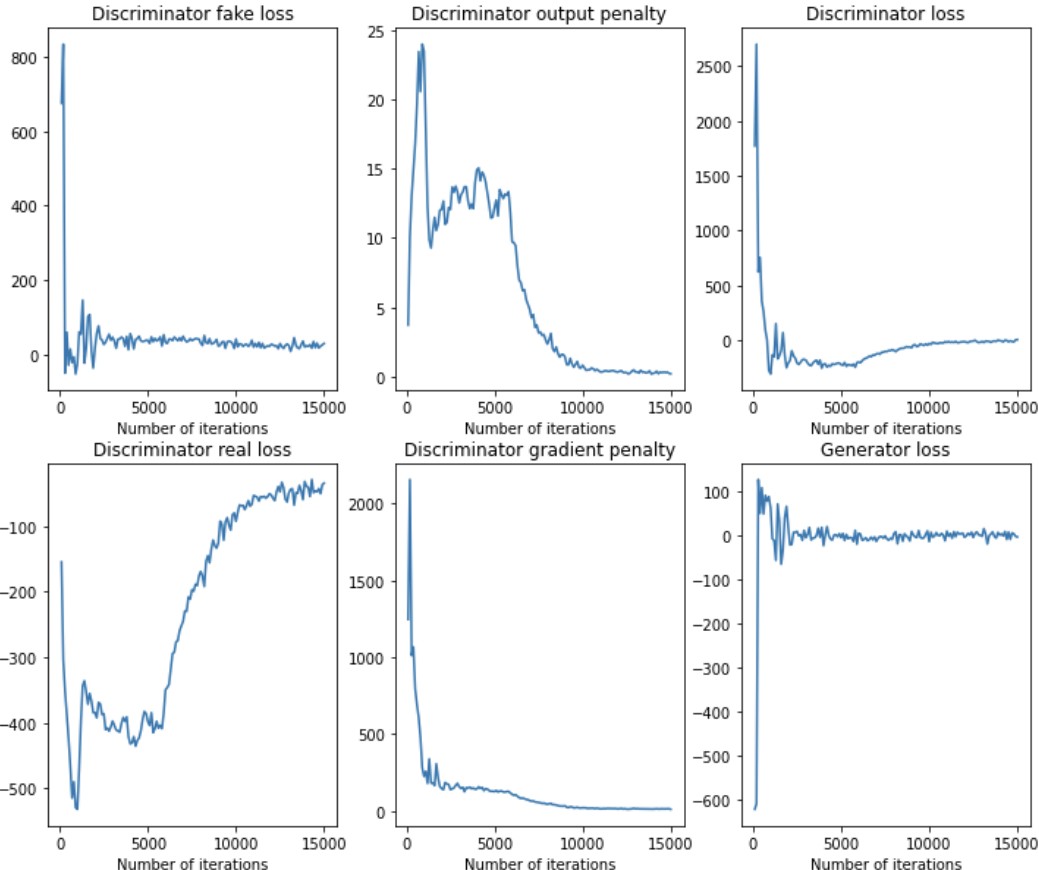

Figure 6: Loss curves of the final $64 \times 64$ stage of training of the non-convex generator and non-concave discriminator as trained with the baseline Progressive GDA method as used in the main paper, for images shown in Figure 3. Discriminator fake loss corresponds to the total network output over the fake images, while real loss corresponds to the negative of the total network output over the real images, output penalty corresponds to the $\epsilon_{\text{drift}}\mathbb{E}_{x\sim p_x}[D(x)^2]$ penalty, gradient penalty refers to the GP loss with $\lambda = 10$, discriminator loss is the sum over all of the discriminator losses, and generator loss corresponds to the negative of the discriminator fake loss.

Table 3: FID results of progressive training of linear generators and two-layer quadratic-activation discriminators using both the convex approach and the non-convex baseline. Results are reported over three runs.

| Method | FID |
|---|---|
| Progressive GDA (Baseline) | $194.1 \pm 4.5$ |
| ProCoGAN (Ours): $(\beta_d^{(4)}, \beta_d^{(5)}) = (7.2 \times 10^3, 1.0 \times 10^4)$ | $128.4 \pm 0.4$ |
| ProCoGAN (Ours): $(\beta_d^{(4)}, \beta_d^{(5)}) = (1.9 \times 10^4, 3.3 \times 10^4)$ | $147.1 \pm 2.4$ |

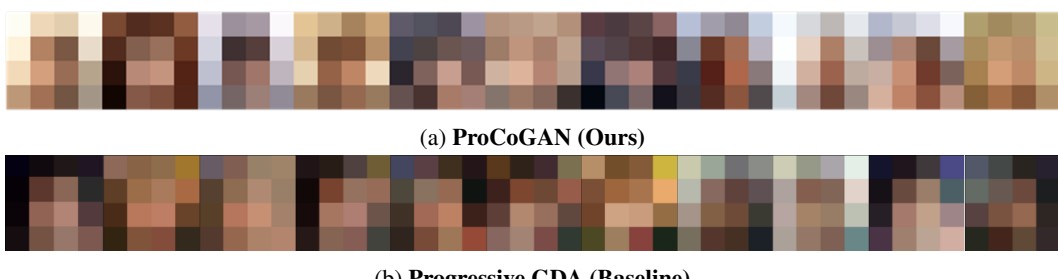

(a) **ProCoGAN (Ours)**

(b) **Progressive GDA (Baseline)**

Figure 7: Representative generated faces at $4 \times 4$ resolution from ProCoGAN and Progressive GDA with stagewise training of *linear* generators and quadratic-activation discriminators on CelebA (Figure 2).

progressively growing the discriminator as well as the generator, using convolutional layers rather than fully-connected layers, using leaky-ReLU activation rather than linear or quadratic-activation, fusing the outputs of different resolutions, and fine-tuning the weights of previous stages when a new stage is being trained. The objective of this experiment is not to replicate (Karras et al., 2017) exactly with a convex algorithm, but rather to simply demonstrate a proof-of-concept for the effectiveness of our equivalent convex program as an alternative to standard GDA applied to the non-concave and non-convex original optimization problem, when both approaches are applied to the same architecture of a linear generator and quadratic-activation two-layer discriminator.

For ProCoGAN, for both of the sets of faces visualized in the main paper, we arbitrarily choose $(\beta_d^{(1)}, \beta_d^{(2)}, \beta_d^{(3)}) = (206, 1.6 \times 10^3, 5.9 \times 10^3)$. $\beta_d^{(i)}$ are in general chosen to truncate $k_i$ singular values of $\mathbf{X}_i = \mathbf{U}_i \mathbf{\Sigma}_i \mathbf{V}_i$, where $k_i$ can be varied.

Both methods are trained with Pytorch (Paszke et al., 2019), where ProCoGAN is trained with a single 12 GB NVIDIA Titan Xp GPU, while progressive GDA is trained with two of them. For numerical results, we use Fréchet Inception Distance (FID) as a metric (Heusel et al., 2017), generated from 1000 generated images from each model compared to the 50000 ground-truth images used for training, reported over three runs. We display our results in Table 3. We find that low values of $\beta_d$ seem to improve the FID metric for ProCoGAN, and these greatly outperform the baseline in terms of FID in both cases. In addition, to show further the progression of the greedy training, for both ProCoGAN and Progressive GDA in the settings described in the main paper, we show representative outputs of each trained generator at each stage of training in Figures 7, 8, 9, and 10.

Further, we ablate the values of $\beta_d^{(i)}$ for ProCoGAN to show in an even more extreme case the tradeoff between smoothness and diversity, and ablate $\beta_g^{(i)}$ in the case of ProgressiveGDA, which provides a similar tradeoff, as we show in Figure 11.

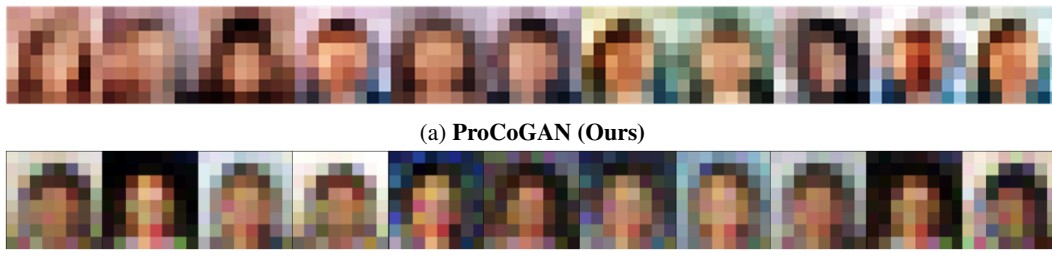

(a) **ProCoGAN (Ours)**

(b) **Progressive GDA (Baseline)**

Figure 8: Representative generated faces at $8 \times 8$ resolution from ProCoGAN and Progressive GDA with stagewise training of *linear* generators and quadratic-activation discriminators on CelebA (Figure 2).

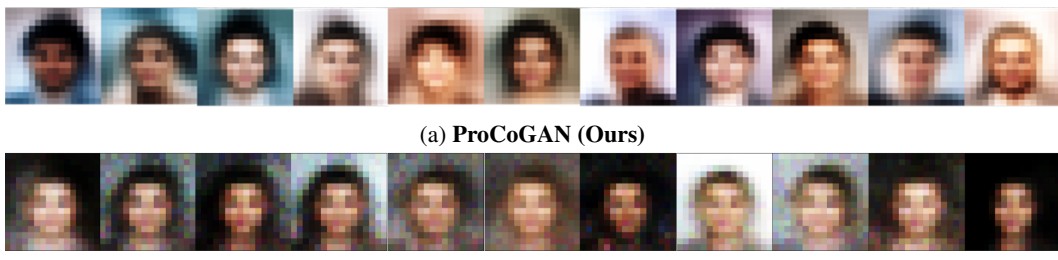

(a) **ProCoGAN (Ours)**

(b) **Progressive GDA (Baseline)**

Figure 9: Representative generated faces at $16 \times 16$ resolution from ProCoGAN and Progressive GDA with stagewise training of *linear* generators and quadratic-activation discriminators on CelebA (Figure 2).

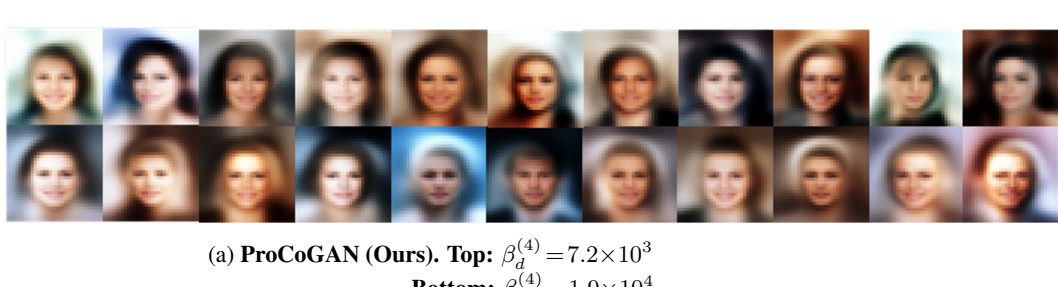

(a) **ProCoGAN (Ours). Top:** $\beta_d^{(4)} = 7.2 \times 10^3$
**Bottom:** $\beta_d^{(4)} = 1.9 \times 10^4$

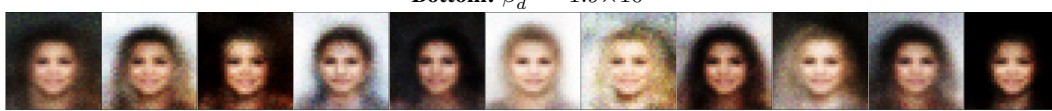

(b) **Progressive GDA (Baseline)**

Figure 10: Representative generated faces at $32 \times 32$ resolution from ProCoGAN and Progressive GDA with stagewise training of *linear* generators and quadratic-activation discriminators on CelebA (Figure 2). ProCoGAN only employs the closed-form expression equation 15, where $\beta_d$ controls the variation and smoothness in the generated images.

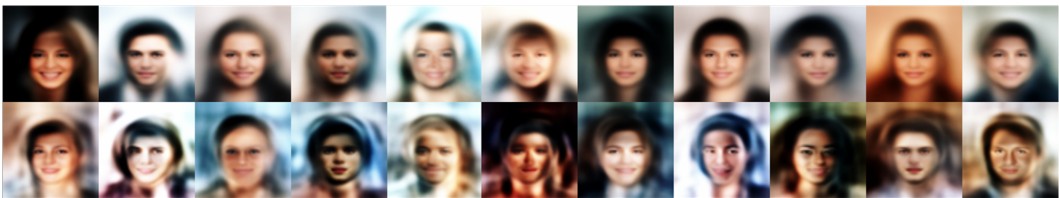

(a) **ProCoGAN (Ours). Top:** $\beta_d^{(i)} = (1.3 \times 10^3, 2.7 \times 10^3, 9.0 \times 10^3, 2.6 \times 10^4, 6.4 \times 10^4)$
**Bottom:** $\beta_d^{(i)} = (51, 557, 2.9 \times 10^3, 5.3 \times 10^3, 6.2 \times 10^3)$

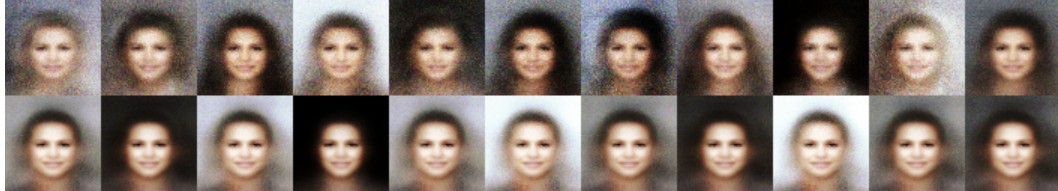

(b) **Progressive GDA (Baseline). Top:** $\beta_g^{(i)} = 10/d_r^{(i)}$
**Bottom:** $\beta_g^{(i)} = 1000/d_r^{(i)}$

Figure 11: Effect of $\beta_d^{(i)}$ on generated faces from ProCoGAN and effect of $\beta_g^{(i)}$ on generated faces from Progressive GDA with stagewise training of *linear* generators and quadratic-activation discriminators on CelebA (Figure 2). ProCoGAN only employs the closed-form expression equation 15, where $\beta_d$ controls the variation and smoothness in the generated images, which can clearly be seen in the extreme example here. We also see that $\beta_g$ has a similar effect for Progressive GDA, where high values of $\beta_g$ make output images less noisy but also less diverse.

## C  ADDITIONAL THEORETICAL RESULTS

### C.1  CONVEXITY AND POLYNOMIAL-TIME TRAINABILITY OF TWO-LAYER RELU GENERATORS

In this section, we re-iterate the results of (Sahiner et al., 2020a) for demonstrating an equivalent convex formulation to the generator problem equation 5:

$$\mathbf{W}_1^*, \mathbf{W}_2^* = \arg\min_{\mathbf{W}_1, \mathbf{W}_2} \|\mathbf{W}_1\|_F^2 + \|\mathbf{W}_2\|_F^2 \text{ s.t. } \mathbf{G}^* = (\mathbf{Z}\mathbf{W}_1)_+\mathbf{W}_2.$$

In the case of ReLU-activation generators, this form appears in many of our results and proofs. Thus, we establish the following Lemma.

**Lemma C.1.** *The non-convex problem equation 5 is equivalent to the following convex optimization problem*

$$\{\mathbf{V}_i^*\}_{i=1}^{|\mathcal{H}_z|} = \arg\min_{\mathbf{V}_i} \sum_{i=1}^{|\mathcal{H}_z|} \|\mathbf{V}_i\|_{\mathrm{K}_i, *} \text{ s.t. } \mathbf{G}^* = \sum_{i=1}^{|\mathcal{H}_z|} \mathbf{H}_z^{(i)} \mathbf{Z} \mathbf{V}_i$$

*for $\|\mathbf{V}_i\|_{\mathrm{K}_i, *} := \min_{t \geq 0} t$ s.t. $\mathbf{V}_i \in t\text{conv}\{\mathbf{Z} = \mathbf{h}\mathbf{g}^T : (2\mathbf{H}_z^{(i)} - \mathbf{I}_{n_f})\mathbf{Z}\mathbf{u} \geq 0, \|\mathbf{Z}\|_* \leq 1\}$, provided that the number of neurons $m_g \geq n_f d_r + 1$. Further, this problem has complexity $\mathcal{O}(n^{r_f}(\frac{n_f}{d_f})^{3r_f})$, where $r_f := \mathrm{rank}(\mathbf{Z})$.*

*Proof.* We begin by re-writing equation 5 in terms of individual neurons:

$$\min_{\mathbf{u}_j, \mathbf{v}_j} \sum_{j=1}^{m_g} \|\mathbf{u}_j\|_2^2 + \|\mathbf{v}_j\|_2^2 \text{ s.t.} \mathbf{G}^* = \sum_{j=1}^{m_g} (\mathbf{Z}\mathbf{u}_j)_+ \mathbf{v}_j^\top. \tag{20}$$

Then, we can restate the problem equivalently as (see D.1):

$$\min_{\|\mathbf{u}_j\|_2 \leq 1, \mathbf{v}_j} \sum_{j=1}^{m_g} \|\mathbf{v}_j\|_2 \text{ s.t.} \mathbf{G}^* = \sum_{j=1}^{m_g} (\mathbf{Z}\mathbf{u}_j)_+ \mathbf{v}_j^\top. \tag{21}$$

Then, we take the dual of this problem as in (Ergen & Pilanci, 2021a; 2020; Sahiner et al., 2020a). First, form the Lagrangian

$$\min_{\|\mathbf{u}_j\|_2 \leq 1, \mathbf{v}_j} \max_{\mathbf{R}} \sum_{j=1}^{m_g} \|\mathbf{v}_j\|_2 + \mathrm{tr}(\mathbf{R}^\top \mathbf{G}^*) - \sum_{j=1}^{m_g} \mathrm{tr}(\mathbf{R}^\top (\mathbf{Z}\mathbf{u}_j)_+ \mathbf{v}_j^\top). \tag{22}$$

Then, by Sion's minimax theorem, we can exchange the minimum over $\mathbf{v}$ and maximum over $\mathbf{R}$, to obtain

$$\min_{\|\mathbf{u}_j\|_2 \leq 1} \max_{\mathbf{R}} \min_{\mathbf{v}_j} \sum_{j=1}^{m_g} \|\mathbf{v}_j\|_2 + \mathrm{tr}(\mathbf{R}^\top \mathbf{G}^*) - \sum_{j=1}^{m_g} \mathrm{tr}(\mathbf{R}^\top (\mathbf{Z}\mathbf{u}_j)_+ \mathbf{v}_j^\top). \tag{23}$$

Minimizing this over $\mathbf{v}$, we obtain the equivalent problem

$$\min_{\|\mathbf{u}_j\|_2 \leq 1} \max_{\mathbf{R}} \mathrm{tr}(\mathbf{R}^\top \mathbf{G}^*) \text{ s.t.} \|\mathbf{R}^\top (\mathbf{Z}\mathbf{u}_j)_+\|_2 \leq 1 \ \forall j \in [m_g]. \tag{24}$$

Under the condition $m_g \geq n_f d_r + 1$, we obtain the equivalent semi-infinite strong dual problem

$$\max_{\mathbf{R}} \mathrm{tr}(\mathbf{R}^\top \mathbf{G}^*) \text{ s.t.} \|\mathbf{R}^\top (\mathbf{Z}\mathbf{u})_+\|_2 \leq 1 \ \forall \|\mathbf{u}\|_2 \leq 1. \tag{25}$$

This over-parameterization requirement arises from the argument that this semi-infinite dual constraint can be supported by at most $m_g^* \leq n_f d_r + 1$ neurons $\mathbf{u}$, and thus if $m_g \geq m_g^*$, strong duality holds (see Lemma 5 of (Sahiner et al., 2020a), Section 3 of (Shapiro, 2009)). This problem can further be re-written as

$$\max_{\mathbf{R}} \mathrm{tr}(\mathbf{R}^\top \mathbf{G}^*) \text{ s.t.} \max_{\|\mathbf{u}\|_2 \leq 1} \|\mathbf{R}^\top (\mathbf{Z}\mathbf{u})_+\|_2 \leq 1. \tag{26}$$

Using the concept of dual norm, we introduce the variable $\mathbf{w}$ to obtain the equivalent problem

$$\max_{\mathbf{R}} \mathrm{tr}(\mathbf{R}^\top \mathbf{G}^*) \text{ s.t.} \max_{\substack{\|\mathbf{u}\|_2 \leq 1 \\ \|\mathbf{w}\|_2 \leq 1}} \mathbf{w}^\top \mathbf{R}^\top (\mathbf{Z}\mathbf{u})_+ \leq 1. \tag{27}$$

Then, we enumerate over all potential sign patterns to obtain

$$\max_{\mathbf{R}} \mathrm{tr}(\mathbf{R}^\top \mathbf{G}^*) \text{ s.t.} \max_{\substack{\|\mathbf{u}\|_2 \leq 1 \\ \|\mathbf{w}\|_2 \leq 1 \\ i \in [|\mathbf{H}_z^{(i)}|] \\ (2\mathbf{H}_z^{(i)} - \mathbf{I}_{n_f})\mathbf{Z}\mathbf{u} \geq 0}} \mathbf{w}^\top \mathbf{R}^\top \mathbf{H}_z^{(i)} \mathbf{Z}\mathbf{u} \leq 1, \tag{28}$$

which we can equivalently write as

$$\max_{\mathbf{R}} \mathrm{tr}(\mathbf{R}^\top \mathbf{G}^*) \text{ s.t.} \max_{\substack{\|\mathbf{u}\|_2 \leq 1 \\ \|\mathbf{w}\|_2 \leq 1 \\ i \in [|\mathbf{H}_z^{(i)}|] \\ (2\mathbf{H}_z^{(i)} - \mathbf{I}_{n_f})\mathbf{Z}\mathbf{u} \geq 0}} \langle \mathbf{R}, \mathbf{H}_z^{(i)} \mathbf{Z}\mathbf{u}\mathbf{w}^\top \rangle \leq 1, \tag{29}$$

which can further be simplified as

$$\max_{\mathbf{R}} \mathrm{tr}(\mathbf{R}^\top \mathbf{G}^*) \text{ s.t.} \max_{\|\mathbf{V}_i\|_{\mathrm{K}_i, *} \leq 1} \langle \mathbf{R}, \mathbf{H}_z^{(i)} \mathbf{Z}\mathbf{V}_i \rangle \leq 1 \ \forall i \in [|\mathbf{H}_z|]. \tag{30}$$

We note that unlike the set of rank-one matrices satisfying the affine constraint when parameterized by $\mathbf{u}\mathbf{w}^\top$, the matrices $\mathbf{V}_i \in \mathcal{K}_i$ are not necessarily rank-one, and can in fact be full-rank. In particular, $\mathcal{K}_i$ is the convex hull of rank-one matrices for which the left factors satisfy the $i$th affine ReLU constraint. We then take the Lagrangian problem

$$\max_{\mathbf{R}} \min_{\lambda \geq 0} \mathrm{tr}(\mathbf{R}^\top \mathbf{G}^*) + \sum_{i=1}^{|\mathbf{H}_z|} \lambda_i (1 - \max_{\|\mathbf{V}_i\|_{\mathrm{K}_i, *} \leq 1} \langle \mathbf{R}, \mathbf{H}_z^{(i)} \mathbf{Z}\mathbf{V}_i \rangle), \tag{31}$$

or equivalently

$$\max_{\mathbf{R}} \min_{\lambda \geq 0} \min_{\|\mathbf{V}_i\|_{\mathrm{K}_i, *} \leq 1} \mathrm{tr}(\mathbf{R}^\top \mathbf{G}^*) + \sum_{i=1}^{|\mathbf{H}_z|} \lambda_i - \lambda_i \langle \mathbf{R}, \mathbf{H}_z^{(i)} \mathbf{Z}\mathbf{V}_i \rangle. \tag{32}$$

By Sion's minimax theorem, we can change the order of the maximum and minimum. Then, maximizing over $\mathbf{R}$ leads to

$$\min_{\lambda \geq 0} \min_{\|\mathbf{V}_i\|_{\mathrm{K}_i, *} \leq 1} \sum_{i=1}^{|\mathbf{H}_z|} \lambda_i \text{ s.t. } \mathbf{G}^* = \sum_{i=1}^{|\mathbf{H}_z|} \lambda_i \mathbf{H}_z^{(i)} \mathbf{Z} \mathbf{V}_i. \tag{33}$$

Lastly, we note that this is equivalent to

$$\arg\min_{\mathbf{V}_i} \sum_{i=1}^{|\mathbf{H}_z|} \|\mathbf{V}_i\|_{\mathrm{K}_i, *} \text{ s.t. } \mathbf{G}^* = \sum_{i=1}^{|\mathbf{H}_z|} \mathbf{H}_z^{(i)} \mathbf{Z} \mathbf{V}_i \tag{34}$$

as desired. The computational complexity of this problem is given as $\mathcal{O}(n^{r_f}(\frac{n_f}{d_f})^{3r_f})$, where $r_f := \mathrm{rank}(\mathbf{Z})$ (see Table 1 of (Sahiner et al., 2020a)). The general intuition behind this complexity is that this problem can be solved with a Frank-Wolfe algorithm, each step of which requires the solution to $|\mathcal{H}_z| \leq \mathcal{O}(r_f(n^{r_f}/d_f)^{r_f})$ subproblems, each of which has complexity $\mathcal{O}(n^{r_f})$. To obtain the weights to the original problem equation 5, we factor $\mathbf{V}_i^* = \sum_{j=1}^{d_r} \mathbf{h}_{ij}^* \mathbf{g}_{ij}^*$ where $(2\mathbf{H}_z^{(i)} - \mathbf{I}_{n_f})\mathbf{Z}\mathbf{h}_{ij}^* \geq 0$ and $\|\mathbf{g}_{ij}^*\|_2 = 1$, and then form

$$(\mathbf{w}_{1ij}^*, \mathbf{w}_{2ij}^*) = \left( \frac{\mathbf{h}_{ij}^*}{\sqrt{\|\mathbf{h}_{ij}^*\|_2}}, \mathbf{g}_{ij}^* \sqrt{\|\mathbf{h}_{ij}^*\|_2} \right), \; i \in [|\mathcal{H}_z|], \; j \in [d_r]$$

as the $ij$th row of $\mathbf{W}_1^*$ and $ij$th column of $\mathbf{W}_2^*$, respectively. Re-substituting these into equation 5 obtains a feasible point with the same objective as the equivalent convex program equation 6. $\qquad\square$

## C.2 NORM-CONSTRAINED DISCRIMINATOR DUALITY

In this section, we consider the discriminator duality results in light of weight norm constraints, rather than regularization, and find that many of the same conclusions hold. In order to model a 1-Lipschitz constraint, we can use the constraint $\{\sum_j |v_j| \leq 1, \|\mathbf{u}_j\|_2 \leq 1\}$. Then, for a linear-activation discriminator, for any data samples $\mathbf{a}, \mathbf{b}$, we have

$$|\sum_{j=1}^m \mathbf{a}^\top \mathbf{u}_j v_j - \sum_{j=1}^m \mathbf{b}^\top \mathbf{u}_j v_j| = |\sum_{j=1}^m \left[ \mathbf{a}^\top \mathbf{u}_j - \mathbf{b}^\top \mathbf{u}_j \right] v_j|$$

$$\leq \max_{\|\mathbf{u}_j\|_2 \leq 1} \left[ \mathbf{a}^\top \mathbf{u}_j - \mathbf{b}^\top \mathbf{u}_j \right]$$

$$= \|\mathbf{a} - \mathbf{b}\|_2.$$

Thus, $\{\sum_j |v_j| \leq 1, \|\mathbf{u}_j\|_2 \leq 1\}$ implies 1-Lipschitz for linear-activation discriminators. For discriminators with other activation functions, we use the same set of constraints as well.

**Lemma C.2.** *A WGAN problem with norm-constrained two-layer discriminator, of the form*

$$p^* = \min_{\theta_g} \max_{\sum_j |v_j| \leq 1, \|\mathbf{u}_j\|_2 \leq 1} \sum_{j=1}^m \left[ \mathbf{1}^\top \sigma(\mathbf{X}\mathbf{u}_j) - \mathbf{1}^\top \sigma(G_{\theta_g}(\mathbf{Z})\mathbf{u}_j) \right] v_j + \mathcal{R}_g(\theta_g)$$

*with arbitrary non-linearity $\sigma$, can be expressed as the following:*

$$p^* = \min_{\theta_g} \max_{\|\mathbf{u}\|_2 \leq 1} \left| \mathbf{1}^\top \sigma(\mathbf{X}\mathbf{u}) - \mathbf{1}^\top \sigma(G_{\theta_g}(\mathbf{Z}\mathbf{u})) \right| + \mathcal{R}_g(\theta_g)$$

*Proof.* We first note that by the definition of the dual norm, we have

$$\max_{\sum_j |v_j| \leq 1} \sum_{j=1}^m c_j v_j = \max_{\|\mathbf{v}\|_1 \leq 1} \mathbf{c}^T \mathbf{v} = \|\mathbf{c}\|_\infty = \max_{j \in [m]} |c_j|.$$

Using this observation, we can simply maximize with respect to $v_j$ to obtain

$$p^* = \min_{\theta_g} \max_{j \in [m], \|\mathbf{u}_j\|_2 \leq 1} \left| \mathbf{1}^\top \sigma(\mathbf{X}\mathbf{u}_j) - \mathbf{1}^\top \sigma(G_{\theta_g}(\mathbf{Z})\mathbf{u}_j) \right| + \mathcal{R}_g(\theta_g)$$

which we can then re-write as

$$p^* = \min_{\theta_g} \max_{\|\mathbf{u}\|_2 \leq 1} \left| \mathbf{1}^\top \sigma(\mathbf{X}\mathbf{u}) - \mathbf{1}^\top \sigma(G_{\theta_g}(\mathbf{Z})\mathbf{u}) \right| + \mathcal{R}_g(\theta_g)$$

as desired. □

**Corollary C.1.** *A WGAN problem with norm-constrained two-layer discriminator with linear activations $\sigma(t) = t$ can be expressed as the following:*

$$p^* = \min_{\theta_g} \|\mathbf{1}^\top \mathbf{X} - \mathbf{1}^\top G_{\theta_g}(\mathbf{Z})\|_2 + \mathcal{R}_g(\theta_g).$$

*Proof.* Start with the following

$$p^* = \min_{\theta_g} \max_{\|\mathbf{u}\|_2 \leq 1} \left| \mathbf{1}^\top \mathbf{X}\mathbf{u} - \mathbf{1}^\top G_{\theta_g}(\mathbf{Z})\mathbf{u} \right| + \mathcal{R}_g(\theta_g).$$

Solving over the maximization with respect to $\mathbf{u}$ obtains the desired result:

$$p^* = \min_{\theta_g} \|\mathbf{1}^\top \mathbf{X} - \mathbf{1}^\top G_{\theta_g}(\mathbf{Z})\|_2 + \mathcal{R}_g(\theta_g).$$

□

**Corollary C.2.** *A WGAN problem with norm-constrained two-layer discriminator with quadratic activations $\sigma(t) = t^2$ can be expressed as the following:*

$$p^* = \min_{\theta_g} \|\mathbf{X}^\top \mathbf{X} - G_{\theta_g}(\mathbf{Z})^\top G_{\theta_g}(\mathbf{Z})\|_2 + \mathcal{R}_g(\theta_g).$$

*Proof.* Start with the following

$$p^* = \min_{\theta_g} \max_{\|\mathbf{u}\|_2 \leq 1} \left| \mathbf{1}^\top (\mathbf{X}\mathbf{u})^2 - \mathbf{1}^\top (G_{\theta_g}(\mathbf{Z})\mathbf{u})^2 \right| + \mathcal{R}_g(\theta_g),$$

which we can re-write as

$$p^* = \min_{\theta_g} \max_{\|\mathbf{u}\|_2 \leq 1} \left| \mathbf{u}^\top \left[ \mathbf{X}^\top \mathbf{X} - G_{\theta_g}(\mathbf{Z})^\top G_{\theta_g}(\mathbf{Z}) \right] \mathbf{u} \right| + \mathcal{R}_g(\theta_g).$$

Solving the maximization over $\mathbf{u}$ obtains the desired result

$$p^* = \min_{\theta_g} \|\mathbf{X}^\top \mathbf{X} - G_{\theta_g}(\mathbf{Z})^\top G_{\theta_g}(\mathbf{Z})\|_2 + \mathcal{R}_g(\theta_g).$$

□

**Corollary C.3.** *A WGAN problem with norm-constrained two-layer discriminator with ReLU activations $\sigma(t) = (t)_+$ can be expressed as the following:*

$$p^* = \min_{\theta_g} \max_{\substack{j_1 \in [|\mathcal{H}_x|] \\ j_w \in [|\mathcal{H}_g|] \\ \|\mathbf{u}\|_2 \leq 1 \\ (2\mathbf{H}_x^{(j_1)} - \mathbf{I}_{n_r})\mathbf{X}\mathbf{u} \geq 0 \\ (2\mathbf{H}_g^{(j_2)} - \mathbf{I}_{n_f})G_{\theta_g}(\mathbf{Z})\mathbf{u} \geq 0}} \left| \mathbf{1}^\top \mathbf{H}_x^{(j_1)} \mathbf{X}\mathbf{u} - \mathbf{1}^\top \mathbf{H}_g^{(j_2)} G_{\theta_g}(\mathbf{Z})\mathbf{u} \right| + \mathcal{R}_g(\theta_g).$$

*Proof.* We start with

$$p^* = \min_{\theta_g} \max_{\|\mathbf{u}\|_2 \leq 1} \left| \mathbf{1}^\top (\mathbf{X}\mathbf{u})_+ - \mathbf{1}^\top (G_{\theta_g}(\mathbf{Z})\mathbf{u})_+ \right| + \mathcal{R}_g(\theta_g).$$

Now, introducing sign patterns of the real data and generated data, we have

$$p^* = \min_{\theta_g} \max_{\substack{j_1 \in [|\mathcal{H}_x|] \\ j_2 \in [|\mathcal{H}_g|] \\ \|\mathbf{u}\|_2 \leq 1 \\ (2\mathbf{H}_x^{(j_1)} - \mathbf{I}_{n_r})\mathbf{X}\mathbf{u} \geq 0 \\ (2\mathbf{H}_g^{(j_2)} - \mathbf{I}_{n_f})G_{\theta_g}(\mathbf{Z})\mathbf{u} \geq 0}} \left| \mathbf{1}^\top \mathbf{H}_x^{(j_1)} \mathbf{X}\mathbf{u} - \mathbf{1}^\top \mathbf{H}_g^{(j_2)} G_{\theta_g}(\mathbf{Z})\mathbf{u} \right| + \mathcal{R}_g(\theta_g)$$

as desired. □

### C.3 Generator Parameterization for Norm-Constrained Discriminators

Throughout this section, we utilize the norm constrained discriminators detailed in Section C.2.

#### C.3.1 Linear Generator ($\sigma(t) = t$)

**Linear-activation discriminator**. For a linear generator and linear-activation norm-constrained discriminator (see Corollary C.1 for details), we have

$$p^* = \min_{\mathbf{W}} \max_{\|\mathbf{u}\|_2 \leq 1} \left(\mathbf{1}^\top \mathbf{X} - \mathbf{1}^\top \mathbf{Z}\mathbf{W}\right)\mathbf{u} + \mathcal{R}_g(\mathbf{W})$$

$$= \min_{\mathbf{W}} \|\mathbf{1}^\top \mathbf{X} - \mathbf{1}^\top \mathbf{Z}\mathbf{W}\|_2 + \mathcal{R}_g(\mathbf{W}).$$

For arbitrary choice of convex regularizer $\mathcal{R}_g(\mathbf{W})$, this problem is convex.

**Quadratic-activation discriminator ($\sigma(t) = t^2$)**. For a linear generator and quadratic-activation norm-constrained discriminator (see Corollary C.2 for details), we have

$$p^* = \min_{\mathbf{W}} \|\mathbf{X}^\top \mathbf{X} - (\mathbf{Z}\mathbf{W})^\top \mathbf{Z}\mathbf{W}\|_2 + \mathcal{R}_g(\mathbf{W}). \tag{35}$$

If $\text{rank}(\mathbf{Z}) \geq \text{rank}(\mathbf{X})$, with appropriate choice of $\mathcal{R}_g(\mathbf{W}) = \beta_g \|\mathbf{Z}\mathbf{W}\|_F^2$, we can write this as

$$p^* = \min_{\mathbf{G}} \|\mathbf{X}^\top \mathbf{X} - \mathbf{G}\|_2 + \beta_g \|\mathbf{G}\|_*, \tag{36}$$

which is convex. With a symmetric solution $\mathbf{G}^*$ to the above, we can factor it into $\mathbf{G}^* = \mathbf{H}^\top \mathbf{H}$, and solve the system $\mathbf{H} = \mathbf{Z}\mathbf{W}^*$ to find the optimal original generator weight $\mathbf{W}^*$, which when substituted into the original objective in equation 35 will obtain the same objective value. We note that if $\text{rank}(\mathbf{X}) > \text{rank}(\mathbf{Z})$, a valid solution $\mathbf{W}^*$ is not guaranteed because the linear system $\mathbf{H} = \mathbf{Z}\mathbf{W}^*$ has no solutions if $\text{rank}(\mathbf{H}) > \text{rank}(\mathbf{Z})$. However, since $\text{rank}(\mathbf{H}) \leq \text{rank}(\mathbf{X})$, as long as $\text{rank}(\mathbf{Z}) \geq \text{rank}(\mathbf{X})$, we will be able to exactly find original weights $\mathbf{W}^*$ from $\mathbf{G}^*$, and the two problems are equivalent.

**ReLU-activation discriminator ($\sigma(t) = (t)_+$).** For a linear generator and ReLU-activation norm-constrained discriminator (see Corollary C.3 for details), we have

$$p^* = \min_{\mathbf{W}} \max_{\substack{j_1 \in [|\mathcal{H}_x|] \\ j_w \in [|\mathcal{H}_g|] \\ \|\mathbf{u}\|_2 \leq 1 \\ \left(2\mathbf{H}_x^{(j_1)} - \mathbf{I}_{n_r}\right)\mathbf{X}\mathbf{u} \geq 0 \\ \left(2\mathbf{H}_g^{(j_2)} - \mathbf{I}_{n_f}\right)\mathbf{Z}\mathbf{W}\mathbf{u} \geq 0}} \left|\mathbf{1}^\top \mathbf{H}_x^{(j_1)} \mathbf{X}\mathbf{u} - \mathbf{1}^\top \mathbf{H}_g^{(j_2)} \mathbf{Z}\mathbf{W}\mathbf{u} + \mathcal{R}_g(\mathbf{W})\right|$$

For arbitrary choice of convex regularizer $\mathcal{R}_g(\mathbf{W})$, this is a convex-concave problem with coupled constraints, as in the weight-decay penalized case.

#### C.3.2 Polynomial-activation Generator

All of the results of the linear generator section hold, with lifted features (see proof of Theorem 4.2).

#### C.3.3 ReLU-activation Generator

**Linear-activation discriminator ($\sigma(t) = t$)**. With standard weight decay, we have

$$p^* = \min_{\mathbf{W}_1, \mathbf{W}_2} \|\mathbf{1}^\top \mathbf{X} - \mathbf{1}^\top (\mathbf{Z}\mathbf{W}_1)_+ \mathbf{W}_2\|_2 + \frac{\beta_g}{2}\left(\|\mathbf{W}_1\|_F^2 + \|\mathbf{W}_2\|_F^2\right).$$

We can write this as a convex program as follows. For the output of the network $(\mathbf{Z}\mathbf{W}_1)_+ \mathbf{W}_2$, the fitting term is a convex loss function. From (Sahiner et al., 2020a), we know that this is equivalent to the following convex optimization problem

$$p^* = \min_{\mathbf{V}_i} \|\mathbf{1}^\top \mathbf{X} - \mathbf{1}^\top \sum_{i=1}^{|\mathcal{H}_z|} \mathbf{H}_z^{(i)} \mathbf{Z}\mathbf{V}_i\|_2 + \beta_g \sum_{i=1}^{|\mathcal{H}_z|} \|\mathbf{V}_i\|_{\mathrm{K}_i, *},$$

where $\mathcal{K}_i := \mathrm{conv}\{\mathbf{u}\mathbf{g}^\top : (2\mathbf{H}_z^{(i)} - \mathbf{I}_{n_f})\mathbf{Z}\mathbf{u} \geq 0, \|\mathbf{g}\|_2 \leq 1\}$ and $\|\mathbf{V}_i\|_{\mathrm{K},*} := \min_{t \geq 0} t$ s.t. $\mathbf{V}_i \in t\mathcal{K}_i$.

**Quadratic-activation discriminator** ($\sigma(t) = t^2$). We have

$$p^* = \min_{\mathbf{W}_1, \mathbf{W}_2} \|\mathbf{X}^\top\mathbf{X} - ((\mathbf{Z}\mathbf{W}_1)_+\mathbf{W}_2)^\top(\mathbf{Z}\mathbf{W}_1)_+\mathbf{W}_2\|_2 + \mathcal{R}_g(\mathbf{W}_1, \mathbf{W}_2).$$

For appropriate choice of regularizer $\mathcal{R}_g(\mathbf{W}_1, \mathbf{W}_2) = \frac{\beta_g}{2}\|(\mathbf{Z}\mathbf{W}_1)_+\mathbf{W}_2\|_F^2$ and $m_g \geq n_f d_r + 1$, we can write this as

$$\mathbf{G}^* = \arg\min_{\mathbf{W}_1, \mathbf{W}_2} \|\mathbf{X}^\top\mathbf{X} - \mathbf{G}^\top\mathbf{G}\|_2 + \frac{\beta_g}{2}\|\mathbf{G}\|_F^2$$

$$\mathbf{W}_1^*, \mathbf{W}_2^* = \arg\min_{\mathbf{W}_1, \mathbf{W}_2} \|\mathbf{W}_1\|_F^2 + \|\mathbf{W}_2\|_F^2 \text{ s.t. } \mathbf{G}^* = (\mathbf{Z}\mathbf{W}_1)_+\mathbf{W}_2.$$

The latter of which we can re-write in convex form as shown in Lemma C.1:

$$\{\mathbf{V}_i^*\}_{i=1}^{|\mathcal{H}_z|} = \arg\min_{\mathbf{V}_i} \sum_{i=1}^{|\mathcal{H}_z|} \|\mathbf{V}_i\|_{\mathrm{K},*} \text{ s.t. } \mathbf{G}^* = \sum_{i=1}^{|\mathcal{H}_z|} \mathbf{H}_z^{(i)}\mathbf{Z}\mathbf{V}_i$$

for convex sets $\mathcal{K}_i := \mathrm{conv}\{\mathbf{u}\mathbf{g}^\top : (2\mathbf{H}_z^{(i)} - \mathbf{I}_{n_f})\mathbf{Z}\mathbf{u} \geq 0, \|\mathbf{g}\|_2 \leq 1\}$, and $\|\mathbf{V}_i\|_{\mathrm{K},*} := \min_{t \geq 0} t$ s.t. $\mathbf{V}_i \in t\mathcal{K}_i$. Thus, the quadratic-activation discriminator, ReLU-activation generator problem in the case of a norm-constrained discriminator can be written as two convex optimization problems, with polynomial time trainability for $\mathbf{Z}$ of a fixed rank.

**ReLU-activation discriminator** ($\sigma(t) = (t)_+$). In this case, we have

$$\arg\min_{\mathbf{W}_1, \mathbf{W}_2} \max_{\substack{j_1 \in [|\mathcal{H}_x|] \\ j_w \in [|\mathcal{H}_g|] \\ \|\mathbf{u}\|_2 \leq 1 \\ (2\mathbf{H}_x^{(j_1)} - \mathbf{I}_{n_r})\mathbf{X}\mathbf{u} \geq 0 \\ (2\mathbf{H}_g^{(j_2)} - \mathbf{I}_{n_f})(\mathbf{Z}\mathbf{W}_1)_+\mathbf{W}_2\mathbf{u} \geq 0}} \left|\mathbf{1}^\top\mathbf{H}_x^{(j_1)}\mathbf{X}\mathbf{u} - \mathbf{1}^\top\mathbf{H}_g^{(j_2)}(\mathbf{Z}\mathbf{W}_1)_+\mathbf{W}_2\mathbf{u}\right| + \mathcal{R}_g(\mathbf{W}_1, \mathbf{W}_2).$$

Then, for appropriate choice of $\mathcal{R}_g(\mathbf{W}_1, \mathbf{W}_2) = \frac{\beta_g}{2}\|(\mathbf{Z}\mathbf{W}_1)_+\mathbf{W}_2\|_F^2$, assuming $m_g \geq n_f d_r + 1$, this is equivalent to

$$\mathbf{G}^* = \arg\min_{\mathbf{G}} \max_{\substack{j_1 \in [|\mathcal{H}_x|] \\ j_w \in [|\mathcal{H}_g|] \\ \|\mathbf{u}\|_2 \leq 1 \\ (2\mathbf{H}_x^{(j_1)} - \mathbf{I}_{n_r})\mathbf{X}\mathbf{u} \geq 0 \\ (2\mathbf{H}_g^{(j_2)} - \mathbf{I}_{n_f})\mathbf{G}\mathbf{u} \geq 0}} \left|\mathbf{1}^\top\mathbf{H}_x^{(j_1)}\mathbf{X}\mathbf{u} - \mathbf{1}^\top\mathbf{H}_g^{(j_2)}\mathbf{G}\mathbf{u}\right| + \frac{\beta_g}{2}\|\mathbf{G}\|_F^2$$

$$\mathbf{W}_1^*, \mathbf{W}_2^* = \arg\min_{\mathbf{W}_1, \mathbf{W}_2} \|\mathbf{W}_1\|_F^2 + \|\mathbf{W}_2\|_F^2 \text{ s.t. } \mathbf{G}^* = (\mathbf{Z}\mathbf{W}_1)_+\mathbf{W}_2.$$

The latter of which we can re-write in convex form as shown in Lemma C.1:

$$\{\mathbf{V}_i^*\}_{i=1}^{|\mathcal{H}_z|} = \arg\min_{\mathbf{V}_i} \sum_{i=1}^{|\mathcal{H}_z|} \|\mathbf{V}_i\|_{\mathrm{K},*} \text{ s.t. } \mathbf{G}^* = \sum_{i=1}^{|\mathcal{H}_z|} \mathbf{H}_z^{(i)}\mathbf{Z}\mathbf{V}_i$$

for convex sets $\mathcal{K}_i := \mathrm{conv}\{\mathbf{u}\mathbf{g}^\top : (2\mathbf{H}_z^{(i)} - \mathbf{I}_{n_f})\mathbf{Z}\mathbf{u} \geq 0, \|\mathbf{g}\|_2 \leq 1\}$ and norm $\|\mathbf{V}_i\|_{\mathrm{K},*} := \min_{t \geq 0} t$ s.t. $\mathbf{V}_i \in t\mathcal{K}_i$. Thus, the ReLU-activation discriminator, ReLU-activation generator problem in the case of a norm-constrained discriminator can be written as a convex-concave game in sequence with a convex optimization problem.

## D   OVERVIEW OF MAIN RESULTS

### D.1   DERIVATION OF THE FORM IN EQUATION 3

Let us consider a positively homogeneous activation function of degree one, i.e., $\sigma(tx) = t\sigma(x), \forall t \in \mathbb{R}_+$. Note that commonly used activation functions such as linear and ReLU satisfy this assumption. Then, weight decay regularized training problem can be written as

$$p^* = \min_{\theta_g} \max_{\theta_d} \sum_{j=1}^m \left(\mathbf{1}^\top\sigma(\mathbf{X}\mathbf{u}_j) - \mathbf{1}^\top\sigma(G_{\theta_g}(\mathbf{Z})\mathbf{u}_j)\right) v_j + \mathcal{R}_g(\theta_g) - \beta_d \sum_{j=1}^m (\|\mathbf{u}_j\|_2^2 + v_j^2).$$

Then, we first note scaling the discriminator parameters as $\bar{\mathbf{u}}_j = \alpha_j \mathbf{u}_j$ and $\bar{v}_j = v_j/\alpha_j$ does not change the output of the networks as shown below

$$\sum_{j=1}^{m} \sigma(\mathbf{X}\bar{\mathbf{u}}_j)\bar{v}_j = \sum_{j=1}^{m} \sigma(\mathbf{X}\alpha_j\mathbf{u}_j)\frac{v_j}{\alpha_j} = \sum_{j=1}^{m} \sigma(\mathbf{X}\mathbf{u}_j)v_j$$

$$\sum_{j=1}^{m} \sigma(G_{\theta_g}(\mathbf{Z})\bar{\mathbf{u}}_j)\bar{v}_j = \sum_{j=1}^{m} \sigma(G_{\theta_g}(\mathbf{Z})\alpha_j\mathbf{u}_j)\frac{v_j}{\alpha_j} = \sum_{j=1}^{m} \sigma(G_{\theta_g}(\mathbf{Z})\mathbf{u}_j)v_j.$$

Moreover, we have the following AM-GM inequality for the weight decay regularization

$$\sum_{j=1}^{m}(\|\mathbf{u}_j\|_2^2 + v_j^2) \geq 2\sum_{j=1}^{m}(\|\mathbf{u}_j\|_2|v_j|),$$

where the equality is achieved when the scaling factor is chosen as $\alpha_j = \left(\frac{|v_j|}{\|\mathbf{u}_j\|_2}\right)^{1/2}$. Since the scaling operation does not change the right-hand side of the inequality, we can set $\|\mathbf{u}_j\|_2 = 1, \forall j$. Thus, the right-hand side becomes $\|\mathbf{v}\|_1 = \sum_{j=1}^{m} |v_j|$.

We also note that this result was previously derived for linear (Ergen & Pilanci, 2021e) and ReLU (Pilanci & Ergen, 2020; Ergen & Pilanci, 2021d). Similarly, the extensions to polynomial and quadratic activations were presented in (Bartan & Pilanci, 2021).

### D.2  PROOF OF THEOREM 2.1

**Linear-activation discriminator** ($\sigma(t) = t$). The regularized training problem for two-layer ReLU networks for the generator can be formulated as follows

$$p^* = \min_{\mathbf{W}_1,\mathbf{W}_2} \mathcal{R}_g(\mathbf{W}_1,\mathbf{W}_2) \text{ s.t. } \max_{\|\mathbf{u}\|_2\leq 1} |\mathbf{1}^\top\sigma(\mathbf{X}\mathbf{u}) - \mathbf{1}^\top\sigma((\mathbf{Z}\mathbf{W}_1)_+\mathbf{W}_2)\mathbf{u})| \leq \beta_d$$

$$\overset{\sigma(t)=t}{\Longrightarrow} p^* = \min_{\mathbf{W}_1,\mathbf{W}_2} \mathcal{R}_g(\mathbf{W}_1,\mathbf{W}_2) \text{ s.t. } \|\mathbf{1}^\top\mathbf{X} - \mathbf{1}^\top(\mathbf{Z}\mathbf{W}_1)_+\mathbf{W}_2\|_2 \leq \beta_d.$$

Assume that the network is sufficiently over-parameterized (which we will precisely define below). Then, we can write the problem

$$p^* = \min_{\mathbf{G}} \|\mathbf{G}\|_F^2 \text{ s.t. } \|\mathbf{1}^\top\mathbf{X} - \mathbf{1}^\top\mathbf{G}\|_2 \leq \beta_d,$$

where the solution $\mathbf{G}^*$ is given by a convex program. Then, to find the optimal generator weights, one can solve

$$\min_{\mathbf{W}_1,\mathbf{W}_2} \|\mathbf{W}_1\|_F^2 + \|\mathbf{W}_2\|_F^2 \text{ s.t. } \mathbf{G}^* = (\mathbf{Z}\mathbf{W}_1)_+\mathbf{W}_2, \tag{37}$$

which can be solved as a convex optimization problem in polynomial time for $\mathbf{Z}$ of a fixed rank, as shown in Lemma C.1, given by

$$\{\mathbf{V}_i^*\}_{i=1}^{|\mathcal{H}_z|} = \arg\min_{\mathbf{V}_i} \sum_{i=1}^{|\mathcal{H}_z|} \|\mathbf{V}_i\|_{\mathrm{K}_i,*} \text{ s.t. } \mathbf{G}^* = \sum_{i=1}^{|\mathcal{H}_z|} \mathbf{H}_z^{(i)}\mathbf{Z}\mathbf{V}_i$$

for convex sets $\mathcal{K}_i := \mathrm{conv}\{\mathbf{u}\mathbf{g}^\top : (2\mathbf{H}_z^{(i)} - \mathbf{I}_{n_f})\mathbf{Z}\mathbf{u} \geq 0, \|\mathbf{g}\|_2 \leq 1\}$ and norm $\|\mathbf{V}_i\|_{\mathrm{K}_i,*} := \min_{t\geq 0} t$ s.t. $\mathbf{V}_i \in t\mathcal{K}_i$, provided that the generator has $m_g \geq n_f d_r + 1$ neurons, and we can further find the original optimal generator weights $\mathbf{W}_1^*, \mathbf{W}_2^*$ from this problem.

**Quadratic-activation discriminator** ($\sigma(t) = t^2$). Based on the derivations in Section E.3, we start with the problem

$$p^* = \min_{\mathbf{W}_1,\mathbf{W}_2} \mathcal{R}_g(\mathbf{W}_1,\mathbf{W}_2) \text{ s.t. } \|\mathbf{X}^\top\mathbf{X} - ((\mathbf{Z}\mathbf{W}_1)_+\mathbf{W}_2)^\top(\mathbf{Z}\mathbf{W}_1)_+\mathbf{W}_2)\|_2 \leq \beta_d.$$

Assume that the network is sufficiently over-parameterized (which we will precisely define below). Then, we can write the problem

$$p^* = \min_{\mathbf{G}} \|\mathbf{G}\|_F^2 \text{ s.t. } \|\mathbf{X}^\top\mathbf{X} - \mathbf{G}^\top\mathbf{G}\|_2 \leq \beta_d,$$

where the solution $\mathbf{G}^*$ is given by $\mathbf{G} = \mathbf{L}(\mathbf{\Sigma}^2 - \beta_d\mathbf{I})_+^{1/2}\mathbf{V}^\top$ for any orthogonal matrix $\mathbf{L}$. Then, to find the optimal generator weights, one can solve

$$\min_{\mathbf{W}_1, \mathbf{W}_2} \|\mathbf{W}_1\|_F^2 + \|\mathbf{W}_2\|_F^2 \text{ s.t. } \mathbf{G}^* = (\mathbf{Z}\mathbf{W}_1)_+\mathbf{W}_2, \tag{38}$$

which can be solved as a convex optimization problem in polynomial time for $\mathbf{Z}$ of a fixed rank, as shown in Lemma C.1, given by

$$\{\mathbf{V}_i^*\}_{i=1}^{|\mathcal{H}_z|} = \arg\min_{\mathbf{V}_i} \sum_{i=1}^{|\mathcal{H}_z|} \|\mathbf{V}_i\|_{K_i,*} \text{ s.t. } \mathbf{G}^* = \sum_{i=1}^{|\mathcal{H}_z|} \mathbf{H}_z^{(i)}\mathbf{Z}\mathbf{V}_i$$

for convex sets $\mathcal{K}_i := \operatorname{conv}\{\mathbf{u}\mathbf{g}^\top : (2\mathbf{H}_z^{(i)} - \mathbf{I}_{n_f})\mathbf{Z}\mathbf{u} \geq 0, \|\mathbf{g}\|_2 \leq 1\}$ and norm $\|\mathbf{V}_i\|_{K_i,*} := \min_{t \geq 0} t$ s.t. $\mathbf{V}_i \in t\mathcal{K}_i$, provided that the generator has $m_g \geq n_f d_r + 1$ neurons, and we can further find the original optimal generator weights $\mathbf{W}_1^*, \mathbf{W}_2^*$ from this problem.

**ReLU-activation discriminator** ($\sigma(t) = (t)_+$). We start with the following problem, where the ReLU activations are replaced by their equivalent representations based on hyperplane arrangements (see Section E.5),

$$p^* = \min_{\mathbf{W}_1, \mathbf{W}_2} \mathcal{R}_g(\mathbf{W}_1, \mathbf{W}_2)$$

$$\text{s.t.} \max_{\substack{\|\mathbf{u}\|_2 \leq 1 \\ j_1 \in [|\mathcal{H}_x|] \\ j_2 \in [|\mathcal{H}_g|] \\ (2\mathbf{H}_x^{(j_1)} - \mathbf{I}_{n_r})\mathbf{X}\mathbf{u} \geq 0 \\ (2\mathbf{H}_g^{(j_2)} - \mathbf{I}_{n_f})(\mathbf{Z}\mathbf{W}_1)_+\mathbf{W}_2\mathbf{u} \geq 0}} \left|\left(\mathbf{1}^\top\mathbf{H}_x^{(j_1)}\mathbf{X} - \mathbf{1}^\top\mathbf{H}_g^{(j_2)}(\mathbf{Z}\mathbf{W}_1)_+\mathbf{W}_2\right)\mathbf{u}\right| \leq \beta_d$$

.

Assume that the generator network is sufficiently over-parameterized, with $m_g \geq n_f d_r + 1$ neurons. Then, we can write the problem as

$$\mathbf{G}^* = \arg\min_{\mathbf{G}} \|\mathbf{G}\|_F^2$$

$$\text{s.t.} \max_{\substack{\|\mathbf{u}\|_2 \leq 1 \\ j_1 \in [|\mathcal{H}_x|] \\ j_2 \in [|\mathcal{H}_g|] \\ (2\mathbf{H}_x^{(j_1)} - \mathbf{I}_{n_r})\mathbf{X}\mathbf{u} \geq 0 \\ (2\mathbf{H}_g^{(j_2)} - \mathbf{I}_{n_f})\mathbf{G}\mathbf{u} \geq 0}} \left|\left(\mathbf{1}^\top\mathbf{H}_x^{(j_1)}\mathbf{X} - \mathbf{1}^\top\mathbf{H}_g^{(j_2)}\mathbf{G}\right)\mathbf{u}\right| \leq \beta_d$$

and

$$\min_{\mathbf{W}_1, \mathbf{W}_2} \|\mathbf{W}_1\|_F^2 + \|\mathbf{W}_2\|_F^2 \text{ s.t. } \mathbf{G}^* = (\mathbf{Z}\mathbf{W}_1)_+\mathbf{W}_2$$

the latter of which can be solved as a convex optimization problem in polynomial time for $\mathbf{Z}$ of a fixed rank, as shown in Lemma C.1, given by

$$\{\mathbf{V}_i^*\}_{i=1}^{|\mathcal{H}_z|} = \arg\min_{\mathbf{V}_i} \sum_{i=1}^{|\mathcal{H}_z|} \|\mathbf{V}_i\|_{K_i,*} \text{ s.t. } \mathbf{G}^* = \sum_{i=1}^{|\mathcal{H}_z|} \mathbf{H}_z^{(i)}\mathbf{Z}\mathbf{V}_i$$

for convex sets $\mathcal{K}_i := \operatorname{conv}\{\mathbf{u}\mathbf{g}^\top : (2\mathbf{H}_z^{(i)} - \mathbf{I}_{n_f})\mathbf{Z}\mathbf{u} \geq 0, \|\mathbf{g}\|_2 \leq 1\}$ and norm $\|\mathbf{V}_i\|_{K_i,*} := \min_{t \geq 0} t$ s.t. $\mathbf{V}_i \in t\mathcal{K}_i$, provided that the generator has $m_g \geq n_f d_r + 1$ neurons, and we can further find the original optimal the generator weights $\mathbf{W}_1^*, \mathbf{W}_2^*$ from this problem.

The former problem is a convex-concave problem. We begin with by forming the Lagrangian of the constraints:

$$p^* = \min_{\mathbf{G}} \|\mathbf{G}\|_F^2$$

$$\text{s.t.} \min_{\substack{\alpha_{j_1} \geq 0 \\ \gamma_{j_2} \geq 0 \\ \forall j_1 \in [|\mathcal{H}_x|], j_2 \in [|\mathcal{H}_g|]}} \left\|\left(\mathbf{1}^\top\mathbf{H}_x^{(j_1)}\mathbf{X} - \mathbf{1}^\top\mathbf{H}_g^{(j_2)}\mathbf{G}\right) + \alpha_{j_1}^\top\left(2\mathbf{H}_x^{(j_1)} - \mathbf{I}_{n_r}\right)\mathbf{X} + \gamma_{j_2}^\top\left(2\mathbf{H}_g^{(j_2)} - \mathbf{I}_{n_f}\right)\mathbf{G}\right\|_2 \leq \beta_d$$

$$\min_{\substack{\alpha'_{j_1} \geq 0 \\ \gamma'_{j_2} \geq 0 \\ \forall j_1 \in [|\mathcal{H}_x|], j_2 \in [|\mathcal{H}_g|]}} \left\|-\left(\mathbf{1}^\top\mathbf{H}_x^{(j_1)}\mathbf{X} - \mathbf{1}^\top\mathbf{H}_g^{(j_2)}\mathbf{G}\right) + {\alpha'_{j_1}}^\top\left(2\mathbf{H}_x^{(j_1)} - \mathbf{I}_{n_r}\right)\mathbf{X} + {\gamma'_{j_2}}^\top\left(2\mathbf{H}_g^{(j_2)} - \mathbf{I}_{n_f}\right)\mathbf{G}\right\|_2 \leq \beta_d$$

Then, forming the Lagrangian, we have

$$
p^* = \min_{\mathbf{G}} \max_{\substack{\lambda,\lambda' \geq 0 \\ \alpha_{j_1} \geq 0,\, \alpha'_{j_1} \geq 0, \\ \gamma_{j_2} \geq 0,\, \gamma'_{j_2} \geq 0, \\ \forall j_1 \in [|\mathcal{H}_x|],\, j_2 \in [|\mathcal{H}_g|]}} \|\mathbf{G}\|_F^2
$$

$$
- \sum_{j_1 j_2} \lambda_{j_1 j_2} \left( \beta_d - \left\| \left( \mathbf{1}^\top \mathbf{H}_x^{(j_1)} \mathbf{X} - \mathbf{1}^\top \mathbf{H}_g^{(j_2)} \mathbf{G} \right) + \alpha_{j_1}^\top \left( 2\mathbf{H}_x^{(j_1)} - \mathbf{I}_{n_r} \right) \mathbf{X} + \gamma_{j_2}^\top \left( 2\mathbf{H}_g^{(j_2)} - \mathbf{I}_{n_f} \right) \mathbf{G} \right\|_2 \right)
$$

$$
- \sum_{j_1 j_2} \lambda'_{j_1 j_2} \left( \beta_d - \left\| -\left( \mathbf{1}^\top \mathbf{H}_x^{(j_1)} \mathbf{X} - \mathbf{1}^\top \mathbf{H}_g^{(j_2)} \mathbf{G} \right) + {\alpha'_{j_1}}^\top \left( 2\mathbf{H}_x^{(j_1)} - \mathbf{I}_{n_r} \right) \mathbf{X} + {\gamma'_{j_2}}^\top \left( 2\mathbf{H}_g^{(j_2)} - \mathbf{I}_{n_f} \right) \mathbf{G} \right\|_2 \right)
$$

We can then re-write this as

$$
p^* = \min_{\mathbf{G}} \max_{\substack{\|\mathbf{r}_{j_1 j_2}\|_2 \leq 1,\, \|\mathbf{r}'_{j_1 j_2}\|_2 \leq 1 \\ \lambda,\lambda' \geq 0 \\ \alpha_{j_1} \geq 0,\, \alpha'_{j_1} \geq 0, \\ \gamma_{j_2} \geq 0,\, \gamma'_{j_2} \geq 0, \\ \forall j_1 \in [|\mathcal{H}_x|],\, j_2 \in [|\mathcal{H}_g|]}} \|\mathbf{G}\|_F^2
$$

$$
- \sum_{j_1 j_2} \lambda_{j_1 j_2} \left( \beta_d - \left( \left( \mathbf{1}^\top \mathbf{H}_x^{(j_1)} \mathbf{X} - \mathbf{1}^\top \mathbf{H}_g^{(j_2)} \mathbf{G} \right) + \alpha_{j_1}^\top \left( 2\mathbf{H}_x^{(j_1)} - \mathbf{I}_{n_r} \right) \mathbf{X} + \gamma_{j_2}^\top \left( 2\mathbf{H}_g^{(j_2)} - \mathbf{I}_{n_f} \right) \mathbf{G} \right) \mathbf{r}_{j_1 j_2} \right)
$$

$$
- \sum_{j_1 j_2} \lambda'_{j_1 j_2} \left( \beta_d - \left( -\left( \mathbf{1}^\top \mathbf{H}_x^{(j_1)} \mathbf{X} - \mathbf{1}^\top \mathbf{H}_g^{(j_2)} \mathbf{G} \right) + {\alpha'_{j_1}}^\top \left( 2\mathbf{H}_x^{(j_1)} - \mathbf{I}_{n_r} \right) \mathbf{X} + {\gamma'_{j_2}}^\top \left( 2\mathbf{H}_g^{(j_2)} - \mathbf{I}_{n_f} \right) \mathbf{G} \right) \mathbf{r}'_{j_1 j_2} \right)
$$

maximizing over $\alpha, \alpha', \gamma, \gamma'$, we have

$$
p^* = \min_{\mathbf{G}} \max_{\substack{\|\mathbf{r}_{j_1 j_2}\|_2 \leq 1,\, \|\mathbf{r}'_{j_1 j_2}\|_2 \leq 1 \\ \lambda,\lambda' \geq 0}} \|\mathbf{G}\|_F^2 - \beta_d \sum_{j_1 j_2} (\lambda_{j_1 j_2} + \lambda'_{j_1 j_2}) + \sum_{j_1 j_2} \left( \mathbf{1}^\top \mathbf{H}_x^{(j_1)} \mathbf{X} - \mathbf{1}^\top \mathbf{H}_g^{(j_2)} \mathbf{G} \right) (\lambda_{j_1 j_2} \mathbf{r}_{j_1 j_2} - \lambda'_{j_1 j_2} \mathbf{r}'_{j_1 j_2})
$$

$$
\text{s.t.} (2\mathbf{H}_x^{(j_1)} - \mathbf{I}_n) \mathbf{X} \mathbf{r}_{j_1 j_2} \geq 0,\ (2\mathbf{H}_g^{(j_2)} - \mathbf{I}_n) \mathbf{G} \mathbf{r}_{j_1 j_2} \geq 0,\ (2\mathbf{H}_x^{(j_1)} - \mathbf{I}_n) \mathbf{X} \mathbf{r}'_{j_1 j_2} \geq 0,\ (2\mathbf{H}_g^{(j_2)} - \mathbf{I}_n) \mathbf{G} \mathbf{r}'_{j_1 j_2} \geq 0
$$

We can then re-parameterize this problem by letting $\mathbf{r}_{j_1 j_2} = \lambda_{j_1 j_2} \mathbf{r}_{j_1 j_2}$ and $\mathbf{r}'_{j_1 j_2} = \lambda'_{j_1 j_2} \mathbf{r}'_{j_1 j_2}$ to obtain the final form:

$$
p^* = \min_{\mathbf{G}} \max_{\mathbf{r}_{j_1 j_2}, \mathbf{r}'_{j_1 j_2}} \|\mathbf{G}\|_F^2 - \beta_d \sum_{j_1 j_2} (\|\mathbf{r}_{j_1 j_2}\|_2 + \|\mathbf{r}'_{j_1 j_2}\|_2) + \sum_{j_1 j_2} \left( \mathbf{1}^\top \mathbf{H}_x^{(j_1)} \mathbf{X} - \mathbf{1}^\top \mathbf{H}_g^{(j_2)} \mathbf{G} \right) (\mathbf{r}_{j_1 j_2} - \mathbf{r}'_{j_1 j_2})
$$

$$
\text{s.t.} (2\mathbf{H}_x^{(j_1)} - \mathbf{I}_n) \mathbf{X} \mathbf{r}_{j_1 j_2} \geq 0,\ (2\mathbf{H}_g^{(j_2)} - \mathbf{I}_n) \mathbf{G} \mathbf{r}_{j_1 j_2} \geq 0,\ (2\mathbf{H}_x^{(j_1)} - \mathbf{I}_n) \mathbf{X} \mathbf{r}'_{j_1 j_2} \geq 0,\ (2\mathbf{H}_g^{(j_2)} - \mathbf{I}_n) \mathbf{G} \mathbf{r}'_{j_1 j_2} \geq 0
$$

which is a convex-concave game with coupled constraints, as desired. $\qquad \square$

### D.3 Note on Convex-Concave Games with Coupled Constraints

We consider the following convex-concave game with coupled constraints:

$$
p^* = \min_{\mathbf{G}} \max_{\mathbf{r}_{j_1 j_2}, \mathbf{r}'_{j_1 j_2}} \|\mathbf{G}\|_F^2 - \beta_d \sum_{j_1 j_2} (\|\mathbf{r}_{j_1 j_2}\|_2 + \|\mathbf{r}'_{j_1 j_2}\|_2) + \sum_{j_1 j_2} \left( \mathbf{1}^\top \mathbf{H}_x^{(j_1)} \mathbf{X} - \mathbf{1}^\top \mathbf{H}_g^{(j_2)} \mathbf{G} \right) (\mathbf{r}_{j_1 j_2} - \mathbf{r}'_{j_1 j_2})
$$

$$
\text{s.t.} (2\mathbf{H}_x^{(j_1)} - \mathbf{I}_n) \mathbf{X} \mathbf{r}_{j_1 j_2} \geq 0,\ (2\mathbf{H}_g^{(j_2)} - \mathbf{I}_n) \mathbf{G} \mathbf{r}_{j_1 j_2} \geq 0,\ (2\mathbf{H}_x^{(j_1)} - \mathbf{I}_n) \mathbf{X} \mathbf{r}'_{j_1 j_2} \geq 0,\ (2\mathbf{H}_g^{(j_2)} - \mathbf{I}_n) \mathbf{G} \mathbf{r}'_{j_1 j_2} \geq 0
$$

Here, we say the problem has "coupled constraints" because some of the constraints jointly depend on $\mathbf{G}$ and $\mathbf{r}_{j_1 j_2}, \mathbf{r}'_{j_1 j_2}$. The existence of saddle points for this problem, since the constraint set is not jointly convex in all problem variables, is not known (Žaković & Rustem, 2003).

However, if all the constraints are strictly feasible, then by Slater's condition, we know the Lagrangian of the inner maximum has a saddle point. Therefore, in the case of strict feasibility, we can write the problem as

$$
p^* = \min_{\mathbf{G}} \max_{\mathbf{r}_{j_1 j_2}, \mathbf{r}'_{j_1 j_2}} \min_{\lambda_{j_1 j_2}, \lambda'_{j_1 j_2} \geq 0} \|\mathbf{G}\|_F^2 - \beta_d \sum_{j_1 j_2} (\|\mathbf{r}_{j_1 j_2}\|_2 + \|\mathbf{r}'_{j_1 j_2}\|_2) + \sum_{j_1 j_2} \left( \mathbf{1}^\top \mathbf{H}_x^{(j_1)} \mathbf{X} - \mathbf{1}^\top \mathbf{H}_g^{(j_2)} \mathbf{G} \right) (\mathbf{r}_{j_1 j_2} - \mathbf{r}'_{j_1 j_2})
$$

$$
+ \sum_{j_1 j_2} \lambda_{j_1 j_2}^\top (2\mathbf{H}_g^{(j_2)} - \mathbf{I}_n) \mathbf{G} \mathbf{r}_{j_1 j_2} + \sum_{j_1 j_2} {\lambda'_{j_1 j_2}}^\top (2\mathbf{H}_g^{(j_2)} - \mathbf{I}_n) \mathbf{G} \mathbf{r}'_{j_1 j_2}
$$

$$
\text{s.t.} (2\mathbf{H}_x^{(j_1)} - \mathbf{I}_n) \mathbf{X} \mathbf{r}_{j_1 j_2} \geq 0,\ (2\mathbf{H}_x^{(j_1)} - \mathbf{I}_n) \mathbf{X} \mathbf{r}'_{j_1 j_2} \geq 0
$$

which by Slater's condition is further identical to

$$
p^* = \min_{\lambda_{j_1 j_2}, \lambda'_{j_1 j_2} \geq 0} \left[ \min_{\mathbf{G}} \max_{\mathbf{r}_{j_1 j_2}, \mathbf{r}'_{j_1 j_2}} \|\mathbf{G}\|_F^2 - \beta_d \sum_{j_1 j_2} (\|\mathbf{r}_{j_1 j_2}\|_2 + \|\mathbf{r}'_{j_1 j_2}\|_2) + \sum_{j_1 j_2} \left( \mathbf{1}^\top \mathbf{H}_x^{(j_1)} \mathbf{X} - \mathbf{1}^\top \mathbf{H}_g^{(j_2)} \mathbf{G} \right) (\mathbf{r}_{j_1 j_2} - \mathbf{r}'_{j_1 j_2})
$$

$$
+ \sum_{j_1 j_2} \lambda_{j_1 j_2}^\top (2\mathbf{H}_g^{(j_2)} - \mathbf{I}_n) \mathbf{G} \mathbf{r}_{j_1 j_2} + \sum_{j_1 j_2} {\lambda'_{j_1 j_2}}^\top (2\mathbf{H}_g^{(j_2)} - \mathbf{I}_n) \mathbf{G} \mathbf{r}'_{j_1 j_2} \right]
$$

$$
\text{s.t.} (2\mathbf{H}_x^{(j_1)} - \mathbf{I}_n) \mathbf{X} \mathbf{r}_{j_1 j_2} \geq 0, \ (2\mathbf{H}_x^{(j_1)} - \mathbf{I}_n) \mathbf{X} \mathbf{r}'_{j_1 j_2} \geq 0
$$

For a fixed outer values of $\lambda_{j_1 j_2}$, $\lambda'_{j_1 j_2}$, the inner min-max problem no longer has coupled constraints, and has a convex-concave objective with convex constraints on the inner maximization problem. A solution for the inner min-max problem can provably be found with a primal-dual algorithm (Chambolle & Pock, 2011), and we can tune $\lambda_{j_1 j_2}$, $\lambda'_{j_1 j_2}$ as hyper-parameters to minimize the solution of the primal-dual algorithm, to find the global objective $p^*$.

### D.4 PROOF OF THEOREM 2.2

Let us first write the training problem explicitly as

$$
\min_{\theta_g \in \mathcal{C}_g} \max_{u_j, b_j, v_j \in \mathbb{R}} \mathbf{1}^T \sum_{j=1}^{m_d} \left( (\mathbf{x} u_j + b_j)_+ - \left( G_{\theta_g}(\mathbf{z}) u_j + b_j \right)_+ \right) v_j + \beta_d \sum_{j=1}^{m_d} (u_j^2 + v_j^2) + \mathcal{R}_g(\theta_g).
$$

After scaling, the problem above can be equivalently written as

$$
\min_{\theta_g \in \mathcal{C}_g} \mathcal{R}_g(\theta_g) \text{ s.t. } \max_{|u| \leq 1, b} \left| \mathbf{1}^T (\mathbf{x} u + b)_+ - \mathbf{1}^T \left( G_{\theta_g}(\mathbf{z}) u + b \right)_+ \right| \leq \beta_d.
$$

By the overparameterization assumption, we have $\left( G_{\theta_g}(\mathbf{z}) u + b \right)_+ = (\mathbf{w} u + b)_+$. Hence, the problem reduces to

$$
\min_{\mathbf{w} \in \mathbb{R}^n} \mathcal{R}_g(\mathbf{w}) \text{ s.t. } \max_{|u| \leq 1, b} \left| \mathbf{1}^T (\mathbf{x} u + b)_+ - \mathbf{1}^T (\mathbf{w} u + b)_+ \right| \leq \beta_d. \tag{39}
$$

Now, let us focus on the dual constraint and particularly consider the following case

$$
\max_b \left| \sum_{i \in \mathcal{S}_1} (x_i + b) - \sum_{j \in \mathcal{S}_2} (w_j + b) \right| \leq \beta_d, \text{ s.t. } \begin{array}{l} (x_i + b) \geq 0, \ \forall i \in \mathcal{S}_1, \ (x_l + b) \leq 0, \ \forall l \in \mathcal{S}_1^c \\ (w_j + b) \geq 0, \ \forall j \in \mathcal{S}_2, \ (w_k + b) \leq 0, \ \forall k \in \mathcal{S}_2^c, \end{array} \tag{40}
$$

where we assume $u = 1$ and $\mathcal{S}_1$ and $\mathcal{S}_2$ are a particular set of indices of the data samples with active ReLUs for the data and noise samples, respectively. Also note that $\mathcal{S}_1^c$ and $\mathcal{S}_2^c$ are the corresponding complementary sets, i.e., $\mathcal{S}_1^c = [n] \backslash \mathcal{S}_1$ and $\mathcal{S}_2^c = [n] \backslash \mathcal{S}_2$. Thus, the problem reduces to finding the optimal bias value $b$. We first note that the constraint can be compactly written as

$$
\min \left\{ \min_{l \in \mathcal{S}_1^c} -x_l, \min_{k \in \mathcal{S}_2^c} -w_k \right\} \geq b \geq \max \left\{ \max_{i \in \mathcal{S}_1} -x_i, \max_{j \in \mathcal{S}_2} -w_j \right\}.
$$

Since the objective is linear with respect to $b$, the maximum value is achieved when bias takes the value of either the upper-bound or lower-bound of the constraint above. Therefore, depending on the selected indices in the sets $\mathcal{S}_1$ and $\mathcal{S}_2$, the bias parameter will be either $-x_k$ or $-w_k$ for a certain index $k$. Since the similar analysis also holds for $u = -1$ and the other set of indices, a set of optimal solution in general can be defined as

$$
(u^*, b^*) = (\pm 1, \pm x_k / w_k).
$$

Now, due to the assumption $\beta_d \leq \min_{i,j \in [n]: i \neq j} |x_i - x_j|$, we can assume that $x_1 \leq w_1 \leq x_2 \leq \ldots \leq x_n \leq w_n$ without loss of generality. Note that equation 39 will be infeasible otherwise. Then, based on this observation above, the problem in equation 39 can be equivalently written as

$$
\mathbf{w}^* = \arg\min_{\mathbf{w} \in \mathbb{R}^n} \mathcal{R}_g(\mathbf{w}) \text{ s.t. } \left| \sum_{i=j}^{2n} s_i (\tilde{x}_i - \tilde{x}_j) \right| \leq \beta_d, \ \left| \sum_{i=1}^{j} s_i (\tilde{x}_j - \tilde{x}_i) \right| \leq \beta_d, \forall j \in [2n] \tag{41}
$$

where

$$\tilde{x}_i = \begin{cases} x_{\lfloor \frac{i+1}{2} \rfloor}, & \text{if } i \text{ is odd} \\ w_{\frac{i}{2}}, & \text{if } i \text{ is even} \end{cases}, \quad s_i = \begin{cases} +1, & \text{if } i \text{ is odd} \\ -1, & \text{if } i \text{ is even} \end{cases}, \quad \forall i \in [2n].$$

After solving the convex optimization problem above for $\mathbf{w}$, we need to find a two-layer ReLU network generator to model the optimal solution $\mathbf{w}^*$ as its output. Therefore, we can directly use the equivalent convex formulations for two-layer ReLU networks introduced in (Pilanci & Ergen, 2020). In particular, to obtain the network parameters, we solve the following convex optimization problem

$$\{(\mathbf{u}_i^*, \mathbf{v}_i^*)\}_{i=1}^{|\mathcal{H}_z|} = \arg\min_{\mathbf{u}_i, \mathbf{v}_i \in \mathcal{C}_i} \sum_{i=1}^{|\mathcal{H}_z|} \|\mathbf{u}_i\|_2 + \|\mathbf{v}_i\|_2 \text{ s.t. } \mathbf{w}^* = \sum_{i=1}^{|\mathcal{H}_z|} \mathbf{H}_z^{(i)} \mathbf{Z}(\mathbf{u}_i - \mathbf{v}_i),$$

where $\mathcal{C}_i = \{\mathbf{u} \in \mathbb{R}^{d_f} : (2\mathbf{H}_z^{(i)} - \mathbf{I}_n)\mathbf{Z}\mathbf{u} \geq 0\}$ and we assume that $m_g \geq n + 1$. □

# E  TWO-LAYER DISCRIMINATOR DUALITY

## E.1  PROOF OF LEMMA 3.1

We start with the expression from equation 3

$$p^* = \min_{\theta_g} \max_{v_j, \|\mathbf{u}_j\|_2 \leq 1} \sum_{j=1}^m \left[ \mathbf{1}^\top \sigma(\mathbf{X}\mathbf{u}_j) - \mathbf{1}^\top \sigma(G_{\theta_g}(\mathbf{Z})\mathbf{u}_j) \right] v_j + \mathcal{R}_g(\theta_g) - \beta_d \sum_{j=1}^m |v_j|.$$

We now solve the inner maximization problem with respect to $v_j$, which is equivalent to the minimization of an affine objective with $\ell_1$ penalty:

$$p^* = \min_{\theta_g} \mathcal{R}_g(\theta_g) \text{ s.t. } \max_{\|\mathbf{u}\|_2 \leq 1} |\mathbf{1}^\top \sigma(\mathbf{X}\mathbf{u}) - \mathbf{1}^\top \sigma(G_{\theta_g}(\mathbf{Z})\mathbf{u})| \leq \beta_d.$$

□

## E.2  PROOF OF COROLLARY 3.1

We simply plug in $\sigma(t) = t$ into the expression of equation 8:

$$p^* = \min_{\theta_g} \mathcal{R}_g(\theta_g) \text{ s.t. } \max_{\|\mathbf{u}\|_2 \leq 1} \left| \left( \mathbf{1}^\top \mathbf{X} - \mathbf{1}^\top G_{\theta_g}(\mathbf{Z}) \right) \mathbf{u} \right| \leq \beta_d.$$

Then, one can solve the maximization problem in the constraint, to obtain

$$p^* = \min_{\theta_g} \mathcal{R}_g(\theta_g) \text{ s.t. } \|\mathbf{1}^\top \mathbf{X} - \mathbf{1}^\top G_{\theta_g}(\mathbf{Z})\|_2 \leq \beta_d$$

as desired. □

## E.3  PROOF OF COROLLARY 3.2

We note that for rows of $\mathbf{X}$ given by $\{\mathbf{x}_i\}_{i=1}^{n_r}$,

$$\mathbf{1}^\top (\mathbf{X}\mathbf{u})^2 = \sum_{i=1}^{n_r} (\mathbf{x}_i^\top \mathbf{u})^2 = \sum_{i=1}^{n_r} \mathbf{u}^\top \mathbf{x}_i \mathbf{x}_i^\top \mathbf{u} = \mathbf{u}^\top \mathbf{X}^\top \mathbf{X}\mathbf{u}$$

Then, substituting into equation 8, we have:

$$p^* = \min_{\theta_g} \mathcal{R}_g(\theta_g) \text{ s.t. } \max_{\|\mathbf{u}\|_2 \leq 1} |\mathbf{u}^\top \left( \mathbf{X}^\top \mathbf{X} - G_{\theta_g}(\mathbf{Z})^\top G_{\theta_g}(\mathbf{Z}) \right) \mathbf{u}| \leq \beta_d.$$

Then, solving the inner maximization problem over $\mathbf{u}$, we obtain

$$p^* = \min_{\theta_g} \mathcal{R}_g(\theta_g) \text{ s.t. } \|\mathbf{X}^\top \mathbf{X} - G_{\theta_g}(\mathbf{Z})^\top G_{\theta_g}(\mathbf{Z})\|_2 \leq \beta_d$$

as desired. □

### E.4 PROOF OF COROLLARY 3.3

When there is a linear skip connection, we can write the problem as

$$p^* = \min_{\theta_g} \max_{v_j, \mathbf{w}, \|\mathbf{u}_j\|_2 \leq 1} \sum_{j=1}^m \left[ \mathbf{1}^\top \sigma(\mathbf{X}\mathbf{u}_j) - \mathbf{1}^\top \sigma(G_{\theta_g}(\mathbf{Z})\mathbf{u}_j) \right] v_j + \left( \mathbf{1}^\top \mathbf{X} - \mathbf{1}^\top G_{\theta_g}(\mathbf{Z}) \right) \mathbf{w} + \mathcal{R}_g(\theta_g) - \beta_d \sum_{j=1}^m |v_j|,$$

where $\sigma(t) = t^2$. Solving over $\mathbf{w}$ yields the constraint that $\mathbf{1}^\top \mathbf{X} = \mathbf{1}^\top G_{\theta_g}(\mathbf{Z})$. Then, following through the minimization over $v_j$ as in Lemma 3.1 and substitution of the non-linearity as in 3.3, we obtain the desired result. $\qquad\square$

### E.5 PROOF OF COROLLARY 3.4

We start with the problem equation 8, and substitute the ReLU non-linearity

$$p^* = \min_{\theta_g} \mathcal{R}_g(\theta_g) \text{ s.t. } \max_{\|\mathbf{u}\|_2 \leq 1} |\mathbf{1}^\top (\mathbf{X}\mathbf{u})_+ - \mathbf{1}^\top (G_{\theta_g}(\mathbf{Z})\mathbf{u})_+| \leq \beta_d.$$

Then, we can introduce hyper-plane arrangements as described in Section 1.2 over both $\mathbf{X}$ and $G_{\theta_g}(\mathbf{Z})$ to obtain the desired result.

$$p^* = \min_{\theta_g} \mathcal{R}_g(\theta_g)$$

$$\text{s.t.} \max_{\substack{\|\mathbf{u}\|_2 \leq 1 \\ j_1 \in [|\mathcal{H}_x|] \\ j_2 \in [|\mathcal{H}_g|] \\ \left(2\mathbf{H}_x^{(j_1)} - \mathbf{I}_{n_r}\right)\mathbf{X}\mathbf{u} \geq 0 \\ \left(2\mathbf{H}_g^{(j_2)} - \mathbf{I}_{n_f}\right)G_{\theta_g}(\mathbf{Z})\mathbf{u} \geq 0}} \left| \left( \mathbf{1}^\top \mathbf{H}_x^{(j_1)} \mathbf{X} - \mathbf{1}^\top \mathbf{H}_g^{(j_2)} G_{\theta_g}(\mathbf{Z}) \right) \mathbf{u} \right| \leq \beta_d$$

$\qquad\square$

## F GENERATOR PARAMETERIZATION AND CONVEXITY

### F.1 PROOF OF THEOREM 4.1

We will analyze individual cases of various discriminators in the case of a linear generator.

**Linear-activation discriminator** ($\sigma(t) = t$). We start from the dual problem (see Section E.2 for details):

$$p^* = \min_{\mathbf{W}} \mathcal{R}_g(\mathbf{W}) \text{ s.t. } \max_{\|\mathbf{u}\|_2 \leq 1} \mathbf{1}^\top \sigma(\mathbf{X}\mathbf{u}) - \mathbf{1}^\top \sigma(\mathbf{Z}\mathbf{W}\mathbf{u}) \leq \beta_d$$

$$= \min_{\mathbf{W}} \mathcal{R}_g(\mathbf{W}) \text{ s.t. } \max_{\|\mathbf{u}\|_2 \leq 1} (\mathbf{1}^\top \mathbf{X} - \mathbf{1}^\top \mathbf{Z}\mathbf{W})\mathbf{u} \leq \beta_d$$

$$= \min_{\mathbf{W}} \mathcal{R}_g(\mathbf{W}) \text{ s.t. } \|\mathbf{1}^\top \mathbf{X} - \mathbf{1}^\top \mathbf{Z}\mathbf{W})\|_2 \leq \beta_d.$$

Clearly, the objective and constraints are convex, so the solution can be found via convex optimization. Slater's condition states that a saddle point of the Lagrangian exists, and only under the condition that the constraint is strictly feasible. Given $\beta_d > 0$, as long as $\mathbf{1}^\top \mathbf{Z} \neq 0$, we can choose a $\mathbf{W}$ such that $\mathbf{1}^\top \mathbf{X} = \mathbf{1}^\top \mathbf{Z}\mathbf{W}$, and a saddle point exists. The Lagrangian is given by

$$p^* = \min_{\mathbf{W}} \max_{\lambda \geq 0} \mathcal{R}_g(\mathbf{W}) + \lambda(\|\mathbf{1}^\top \mathbf{X} - \mathbf{1}^\top \mathbf{Z}\mathbf{W}\|_2 - \beta_d).$$

Introducing additional variable $\mathbf{r}$, we have also

$$p^* = \min_{\mathbf{W}} \max_{\substack{\lambda \geq 0 \\ \|\mathbf{r}\|_2 \leq 1}} \mathcal{R}_g(\mathbf{W}) + \lambda\left( (\mathbf{1}^\top \mathbf{X} - \mathbf{1}^\top \mathbf{Z}\mathbf{W})\mathbf{r} - \beta_d \right).$$

Now, $\mathbf{v} = \lambda \mathbf{r}$, where $\lambda = \|\mathbf{v}\|_2$

$$p^* = \min_{\mathbf{W}} \max_{\mathbf{v}} \mathcal{R}_g(\mathbf{W}) + (\mathbf{1}^\top \mathbf{X} - \mathbf{1}^\top \mathbf{Z}\mathbf{W})\mathbf{v} - \beta_d \|\mathbf{v}\|_2.$$

From Slater's condition, we can change the order of min and max without changing the objective, which proves there is a saddle point:

$$p^* = \max_{\mathbf{v}} \min_{\mathbf{W}} \mathcal{R}_g(\mathbf{W}) + (\mathbf{1}^\top \mathbf{X} - \mathbf{1}^\top \mathbf{Z}\mathbf{W})\mathbf{v} - \beta_d\|\mathbf{v}\|_2.$$

The inner problem is convex and depending on choice of $\mathcal{R}_g(\mathbf{W})$ can be solved for $\mathbf{W}^*$ in closed form, and subsequently the outer maximization is convex as well. Thus, for a linear generator and linear-activation discriminator, a saddle point provably exists and can be found via convex optimization.

**Quadratic-activation discriminator** ($\sigma(t) = t^2$). We start from the following dual problem (see Section E.3 for details)

$$p^* = \min_{\mathbf{W}} \mathcal{R}_g(\mathbf{W}) \text{ s.t. } \|\mathbf{X}^\top \mathbf{X} - (\mathbf{Z}\mathbf{W})^\top(\mathbf{Z}\mathbf{W})\|_2 \leq \beta_d.$$

This can be lower bounded as follows:

$$p^* \geq d^* = \min_{\mathbf{G}} \frac{\beta_g}{2}\|\mathbf{G}\|_F^2 \text{ s.t. } \|\mathbf{X}^\top \mathbf{X} - \mathbf{G}^\top \mathbf{G}\|_2 \leq \beta_d. \tag{42}$$

Which can further be written as:

$$d^* = \min_{\tilde{\mathbf{G}}} \frac{\beta_g}{2}\|\tilde{\mathbf{G}}\|_* \text{ s.t. } \|\mathbf{X}^\top \mathbf{X} - \tilde{\mathbf{G}}\|_2 \leq \beta_d.$$

This is a convex optimization problem, with a closed-form solution. In particular, if we let $\mathbf{X}^\top \mathbf{X} = \mathbf{V}\boldsymbol{\Sigma}^2\mathbf{V}^\top$ be the eigenvalue decomposition of the covariance matrix, then the solution to equation 42 is found via singular value thresholding:

$$\mathbf{G}^* = \mathbf{V}(\boldsymbol{\Sigma}^2 - \beta_d\mathbf{I})_+\mathbf{V}^\top.$$

This lower bound is achievable if $\exists \mathbf{W} : (\mathbf{Z}\mathbf{W})^\top(\mathbf{Z}\mathbf{W}) = \mathbf{G}^*$. A solution is achieved by allowing $\mathbf{W} = (\mathbf{Z}^\top \mathbf{Z})^{-1/2}(\boldsymbol{\Sigma}^2 - \beta_d\mathbf{I})_+^{1/2}\mathbf{V}^\top$, where computing $(\mathbf{Z}^\top \mathbf{Z})^{-1/2}$ requires inverting only the first $k$ eigenvalue directions[4], where $k := \max_{k:\sigma_k^2 \geq \beta_d} k$. Thus given that $\text{rank}(\mathbf{Z}) \geq k$, the solution of the linear generator, quadratic-activation discriminator can be achieved in closed-form.

In the case that $\text{rank}(\mathbf{Z}) \geq k + 1$, strict feasibility is obtained, and by Slater's condition a saddle point of the Lagrangian exists. One can form the Lagrangian as follows:

$$p^* = \min_{\mathbf{G}} \max_{\mathbf{R}\succeq 0} \frac{\beta_g}{2}\|\mathbf{G}\|_* + \text{tr}(\mathbf{R}\mathbf{X}^\top \mathbf{X}) - \text{tr}(\mathbf{R}\mathbf{G}) - \beta_d\text{tr}(\mathbf{R}).$$

This is a convex-concave game, and from Slater's condition we can exchange the order of the minimum and maximum without changing the objective:

$$p^* = \max_{\mathbf{R}\succeq 0} \min_{\mathbf{G}} \frac{\beta_g}{2}\|\mathbf{G}\|_* + \text{tr}(\mathbf{R}\mathbf{X}^\top \mathbf{X}) - \text{tr}(\mathbf{R}\mathbf{G}) - \beta_d\text{tr}(\mathbf{R}).$$

**ReLU-activation discriminator** ($\sigma(t) = (t)_+$). We again start from the dual problem (see Section E.5 for details)

$$p^* = \min_{\mathbf{W}} \mathcal{R}_g(\mathbf{W})$$

$$\text{s.t.} \max_{\substack{\|\mathbf{u}\|_2 \leq 1 \\ j_1 \in [|\mathcal{H}_x|] \\ j_2 \in [|\mathcal{H}_g|] \\ (2\mathbf{H}_x^{(j_1)} - \mathbf{I}_{n_r})\mathbf{X}\mathbf{u} \geq 0 \\ (2\mathbf{H}_g^{(j_2)} - \mathbf{I}_{n_f})\mathbf{Z}\mathbf{W}\mathbf{u} \geq 0}} \left|\left(\mathbf{1}^\top \mathbf{H}_x^{(j_1)}\mathbf{X} - \mathbf{1}^\top \mathbf{H}_g^{(j_2)}\mathbf{Z}\mathbf{W}\right)\mathbf{u}\right| \leq \beta_d$$

.

We can follow identical steps of the proof of Theorem 2.1 (see Section D.2), with $\mathbf{Z}\mathbf{W}$ instead of $\mathbf{G}$, obtain

$$p^* = \min_{\mathbf{W}} \max_{\mathbf{r}_{j_1 j_2}, \mathbf{r}'_{j_1 j_2}} \mathcal{R}_g(\mathbf{W}) - \beta_d \sum_{j_1 j_2}(\|\mathbf{r}_{j_1 j_2}\|_2 + \|\mathbf{r}'_{j_1 j_2}\|_2) + \sum_{j_1 j_2}\left(\mathbf{1}^\top \mathbf{H}_x^{(j_1)}\mathbf{X} - \mathbf{1}^\top \mathbf{H}_g^{(j_2)}\mathbf{Z}\mathbf{W}\right)(\mathbf{r}_{j_1 j_2} - \mathbf{r}'_{j_1 j_2})$$

$$\text{s.t.}(2\mathbf{H}_x^{(j_1)} - \mathbf{I}_n)\mathbf{X}\mathbf{r}_{j_1 j_2} \geq 0, \ (2\mathbf{H}_g^{(j_2)} - \mathbf{I}_n)\mathbf{Z}\mathbf{W}\mathbf{r}_{j_1 j_2} \geq 0, \ (2\mathbf{H}_x^{(j_1)} - \mathbf{I}_n)\mathbf{X}\mathbf{r}'_{j_1 j_2} \geq 0, \ (2\mathbf{H}_g^{(j_2)} - \mathbf{I}_n)\mathbf{Z}\mathbf{W}\mathbf{r}'_{j_1 j_2} \geq 0$$

as desired. Thus, as long as $\mathcal{R}_g$ is convex in $\mathbf{W}$, we have a convex-concave game with coupled constraints. □

---

[4]For instance, letting $\mathbf{Z} = \mathbf{Q}\boldsymbol{\Lambda}\mathbf{Q}^\top$, we can use $(\mathbf{Z}^\top \mathbf{Z})^{-1/2} = \mathbf{Q}_{[:k]}\boldsymbol{\Lambda}_{[:k]}^{-1}$, where $\cdot_{[:k]}$ indicates taking the first $k$ columns/diagonal entries respectively.

### F.2 PROOF OF THEOREM 4.2

We note that for a polynomial-activation generator with $m$ neurons and corresponding weights $\mathbf{w}_j^{(1)}$, $\mathbf{w}_j^{(2)}$, for samples $\{\mathbf{z}_i\}_{i=1}^{n_f}$:

$$
\begin{aligned}
G_{\theta_g}(\mathbf{z}_i) &= \sum_{j=1}^{m} \sigma(\mathbf{z}_i^\top \mathbf{w}_j^{(1)}) \mathbf{w}_j^{(2)\top} \\
&= \sum_{j=1}^{m} \left( a(\mathbf{z}_i^\top \mathbf{w}_j^{(1)})^2 + b(\mathbf{z}_i^\top \mathbf{w}_j^{(1)}) + c \right) \mathbf{w}_j^{(2)\top} \\
&= \sum_{j=1}^{m} \left( a \langle \mathbf{z}_i \mathbf{z}_i^\top, \mathbf{w}_j^{(1)} \mathbf{w}_j^{(1)\top} \rangle + b(\mathbf{z}_i^\top \mathbf{w}_j^{(1)}) + c \right) \mathbf{w}_j^{(2)\top} \\
&= \sum_{j=1}^{m} \begin{bmatrix} a\mathrm{vec}(\mathbf{z}_i \mathbf{z}_i^\top) \\ b\mathbf{z}_i \\ c \end{bmatrix}^\top \begin{bmatrix} \mathrm{vec}(\mathbf{w}_j^{(1)} \mathbf{w}_j^{(1)\top}) \mathbf{w}_j^{(2)\top} \\ \mathbf{w}_j^{(1)} \mathbf{w}_j^{(2)\top} \\ \mathbf{w}_j^{(2)\top} \end{bmatrix} \\
&= \sum_{j=1}^{m} \tilde{\mathbf{z}}_i^\top \mathbf{w}_j \\
&= \begin{bmatrix} \tilde{\mathbf{z}}_1^\top \\ \tilde{\mathbf{z}}_2^\top \\ \dots \\ \tilde{\mathbf{z}}_{n_f}^\top \end{bmatrix} \mathbf{W} \\
&= \tilde{\mathbf{Z}}\mathbf{W}
\end{aligned}
$$

for $\tilde{\mathbf{z}}_i := \begin{bmatrix} a\mathrm{vec}(\mathbf{z}_i \mathbf{z}_i^\top) \\ b\mathbf{z}_i \\ c \end{bmatrix}$ as the lifted features of the inputs, and a re-parameterized weight matrix $\mathbf{w}_j := \begin{bmatrix} \mathrm{vec}(\mathbf{w}_j^{(1)} \mathbf{w}_j^{(1)\top}) \mathbf{w}_j^{(2)\top} \\ \mathbf{w}_j^{(1)} \mathbf{w}_j^{(2)\top} \\ \mathbf{w}_j^{(2)\top} \end{bmatrix}$ (Bartan & Pilanci, 2021). Thus, any two-layer polynomial-activation generator can be re-parameterized as a linear generator, and thus after substituting $\tilde{\mathbf{Z}}$ as $\mathbf{Z}$ for Theorem 4.1, we can obtain the desired results. $\qquad\square$

