# OpenReview forum: "Hidden Convexity of Wasserstein GANs: Interpretable Generative Models with Closed-Form Solutions"
_ICLR.cc/2022/Conference — ICLR 2022 Poster_

### Official Review · Reviewer_5mJ8 · 2021-10-31

**Correctness:** 3
**Technical Novelty And Significance:** 4
**Empirical Novelty And Significance:** 4
**Recommendation:** 8
**Confidence:** 5

**Main Review:**

This is an interesting paper. I like the idea of reformulating WGAN as a convex problem under a simplified two-layer architecture and then using matrix algebra to solve it analytically.

**Strength**
* Solid theoretical analysis.
* Impressive experimental results

**Weakness**
* Lack of clarity articulating the theory and alogirthm

**Detailed comments**

* Some of the claims made by the author(s) are too strong. To the best of my knowledge, WGANs are only sparsely used in practical applications, yet the author(s) are making the impression that WGANs has become an integral part of standard CV tools, which is not the case. Such examples include:
  * GANs have become arguably the workhorse of computer vision
  * their (GANs') prevalent utilization

* The author(s) claimed that: "For the first time, we show that WGAN can provably be expressed as a convex problem (or a convex-concave game)". I believe this statement is not entirely accurate. WGAN is the dual form of the optimal transport problem, and the entropy-regularized primal formulation is strictly convex and can be efficiently solved with Sinkhorn iterations. Note that this primal formulation is extensively used under the name earth moving distance [1,2].

[1] M Cuturi. Sinkhorn Distances: Lightspeed Computation of Optimal Transport. NIPS 2013

[2] L Chen, et al. Adversarial text generation via feature-mover's distance. NeurIPS 2018

* This paper is about the hidden convexity of WGAN, but the related work section seems to solely focus on the discussion of GANs rather than WGANs. Also, despite claiming "convexity has been seldomly exploited for GANs", a long list of works has been discussed, which appeared contradictory.

* The notations used for (matrix, function) dimensions are confusing. Try to avoid using subscripts for the variables defining dimensions $(n_f, n_r, d_f, d_r)$, just use lower case letters and be clear about what they mean. For functions, do not use the "batch-level" definition (e.g., $D: R^{nxd}->R^n$), because I believe each d-dimension vector is processed independently here. When applied to a tensor, the convention is that such a function will automatically consume the last dimension as inputs and return a reduced output tensor shaped the rest of the dimensions.

* Theorem 2.1 is not clear, the statements are quite vague (e.g., "for appropriate choice of regularizer $R_g$", "a series of convex optimization problems", etc.) It is quite confusing to me what does "in polynomial time in all dimensions for noise inputs $Z$ of a fixed rank" mean?

* It is still not clear from Eqns (4-6) how the optimal generator weights $W_1^*$, $W_2^*$ can be constructed. A related solution only appears in Sec 4.

* Not enough intuitions are provided to help understand the solution. I stuck at Eqns (4-6) for quite a while trying to figure out what they mean intuitively. And some vague heuristic is only provided in a paragraph after that (e.g., "first, it solves for the optimal generator output").

* One major concern with the proposed solution is that the convex / convex-concave game only admits the full-batch optimization, but it does not appeal to the mini-batch optimization. This raises serious concerns about the scalability and practical applicability of the proposed solution.

* The analyses only apply to the simple two-layer neural networks, for both discriminator and generator. Although more sophistication can be achieved by progressively 'stacking' the layers, still these architectural constraints exclude many modern architectures (e.g., attention, ResNet, etc.).  That said, I am still impressed with the results achieved with such simple architectures.

* More details and discussions are needed on the numerical experiments. A few questions on top of my head include: (i) how do you pull-off the full-batch optimization with tens of thousands of images? (ii) matrix factorization is very sensitive to numerical precision, so are you using the standard float32 (which I consider problematic) or the double-precision float64?

* Given the superior efficiency of convexity formulation, I think it is better to be used as an initialization scheme. What are the author's thoughts and can you add some discussions? Maybe further efficiency boosts can be expected using some matrix reduction schemes.

* Finally, I think the result critically depends on the choice of convex regularization parameters (e.g., $\beta_d$), yet there is no discussion on the practical guidelines for its choices.

* The moment matching perspective heavily overlaps the idea of MMD-GAN. An important reference the author(s) should have discussed is [3].

[3] A Genevay, et al. Learning Generative Models with Sinkhorn Divergences. AISTATS 2018

**Summary Of The Paper:**


+++ After Rebuttal +++
I updated my rating to accept after reading the author(s)'s rebuttal and other reviews.
+++++++++++++++++

This paper reformulates WGAN as a convex (or convex-concave) game under restricting neural architectures and convex regularizers. An efficient algorithm based on matrix factorization is proposed to solve the problem analytically.

**Summary Of The Review:**

This is a well-crafted paper, with both strong theory and adequate empirical evidence to support the claims. My suggestion is to refine the presentation to highlight both the intuitions and more practical aspects.

---

### Official Review · Reviewer_RNn1 · 2021-11-01

**Correctness:** 4
**Technical Novelty And Significance:** 2
**Empirical Novelty And Significance:** 2
**Recommendation:** 5
**Confidence:** 4

**Main Review:**

Strengths:
1.  The first optimization related results on  WGAN  as a convex problem (or a convex-concave game) with polynomial-time complexity for two-layer discriminators and two-layer generators under various activation functions.
2.  They uncovered the effects of discriminator activation on data generation through moment matching, where quadratic activation matches the covariance, while ReLU activation amounts to  piecewise mean matching.
3. They found closed-form solutions for WGAN training as singular value thresholding, which provides interpretability.
4. Their experiments demonstrate the interpretability and effectiveness of progressive convex
GAN training for generation of CelebA faces.

Weaknesses:
1. It is unclear how the optimization related results on WGAN really improves our understanding WGAN. To this reviewer, knowing these results does not tell why WGAN really works as a good data generator method for many applications.

**Summary Of The Paper:**

In this paper, the authors analyzed the training of Wasserstein GANs with two-layer neural network
discriminators.  They showed that WGAN can provably be expressed as a convex problem (or
a convex-concave game) with polynomial-time complexity for two-layer discriminators and
two-layer generators under various activation functions.
They uncovered the effects of discriminator activation on data generation through moment matching, where quadratic activation matches the covariance, while ReLU activation amounts to
piecewise mean matching.
They found closed-form solutions for WGAN training as singular value thresholding, which provides interpretability.
Their experiments demonstrate the interpretability and effectiveness of progressive convex
GAN training for generation of CelebA faces.

**Summary Of The Review:**

Although there are some results on WGAN, knowing these results does not lead to any insights on why WGAN really works as a good data generator method for many applications.

---

### Official Review · Reviewer_Hxh6 · 2021-11-02

**Correctness:** 4
**Technical Novelty And Significance:** 3
**Empirical Novelty And Significance:** 3
**Recommendation:** 8
**Confidence:** 3

**Main Review:**

Overall the paper is well presented. The main results of interpreting the two-layer WGAN training problem as convex problems look solid to me. The experiments are well-conducted and the proposed method is consistently evaluated. The experiment results showed the efficiency and efficacy of the proposed method.

Since under the original WGAN settings, the generator is required to be 1-Lip, so a minor concern is that how this would affect the feasibility of the convex problem. In particular, in section B.2, the 1-Lip constraints for linear activation functions are discussed, but not under other settings. I'm concerned about how this 1-Lip condition is guaranteed with other activation functions, and how this would affect the domain feasibility.

**Summary Of The Paper:**

In this paper, novel theoretical results that express two-layer WGAN training as convex-concave games are established. Ample theoretic theorems and propositions, as well as insightful interpretations, are given under various activation function settings, including linear, quadratic, and ReLU. Finally, proof-of-concept experiments are conducted to verify the theoretic results, where the proposed method that utilizes the explicit convex-concave game solutions are shown to generate better visual quality images than traditional non-convex trained results.

The main novelty of the paper comes from the theoretic results of the convex interpretation of the non-trivial WGAN training problem. Novel closed forms are given for certain training settings. The proposed ProCoGAN model based on the theoretic results is also novel.

**Summary Of The Review:**

This paper proposed for the first time the convex-concave game equivalence of 2-layer WGAN model training problem with general activation functions. Experiment results also favor the theoretic results. I would recommend accepting the paper.

---

### Official Review · Reviewer_x4Dn · 2021-11-05

**Correctness:** 4
**Technical Novelty And Significance:** 3
**Empirical Novelty And Significance:** 3
**Recommendation:** 8
**Confidence:** 5

**Main Review:**

I will only provide feedback for the theoretical part of the paper, as I do not have appropriate experience/exposure to the simulation side of GANs. However, as I read through the experimental results, it seems the author(s) are claiming that using only closed form equation (14) they can generate realistic images and can compete with the baseline progressive GDA. Also the architecture in Figure 2 looks interesting, but as I said I cannot judge the validity/novelty of this paper’s experimental results.

1-	Throughout the paper, you mention several times that your results hold when considering the Z matrix to be fixed rank, for example in Theorem 2.1 and discussion after that regarding equation (6). How/where do you justify that (theoretically and/or practically)?
2-   In equation (6) you use convex sets K_i without defining them!
3-	Some parts of Lemma B.1 and especially its proof are not clear to me. As you use this Lemma multiple times in the proofs of other results, please explain the following:
●	Where is the complexity analysis, which you mention at the end of your Lemma, in the proof? How does rank(Z) appear as an exponent in this bound?
●	In the start of page 20, please provide more details on how you can get the dual form as claimed and how you derive the lower bound on generator hidden nodes count. Is this a novel result in this paper, or did you use the result from the mentioned 2 papers?
●	It would be great if you can talk about the geometric properties of the convex sets Ki. For example as rank(Z) varies, what happens to these sets? How do you compare them to the set of all rank-1 matrices (as a subspace)?
●	When you write the Lagrangian in the middle of page 20 (you can add numbers to make referencing  these equations easy), why suddenly max over Ki changes to min? You should provide more details on deriving these Lagrangian in the paper.
●	Typo?: Then, “minimizing” over R leads to …. Is it maximizing or minimizing?
●	Typo: ijth column and row. Should be something like (i,j)th element of…
4-	At the bottom of page 22, what do you mean by appropriate choice of regularizer? Which choice do you use?
5-	The derivation at the bottom of page 22 requires more justification. How the 2 min problems are equivalent under the condition rank(Z)>= rank(X)
6-	In deriving equation (3) in section C.1, it seems to me that actually by replacing weight decay regularization with 1-norm of v, as justified by AM-GM, we get a upper bound to p*. Is this correct? If not, please explain how we get the exact same min-max by using AM-GM (the scaling part is understood).
7-	Typo: after 3rd equation in page25, as well as page 26: the original optimal “the” generator weights.
8-	In C.2, for the proofs, you should polish/remove/reorganize some arguments like “which we will precisely define below”
9-	In the proofs of C.2, how do you suddenly replace objective Rg(W1,W2) with ||G||F ? Is it related to the assumption of overparameterizing the generator? I do not see any derivation of this sudden replacement which retains the same p*.
10-	Error in dual variables of Lagrangian in page 26: It seems to me that {j}1{j}2 should not be used twice, as you are dealing with two different sets of constraints. I mean instead of 2 sets of dual vars, I think you should use 4; however it doesn’t seem to affect the final conclusion at page 27, and you implicitly correct it at the beginning of page 27.
11-	I would like to get more explanation/insight/intuition about the statement at the end of page 7, “The effect of a polynomial-activation generator is thus to provide more heavy-tailed noise as input to the generator, which provides more degrees of freedom to the generator for modeling more complex data distributions.”
12-	Is there any high-level justification for the claim after equation (14) in page 7 where you say : with p∗ = d∗ under the condition that rank(Z) is sufficiently large
13-	 Equations (14) and (25) do not match! (14) seems to be the correct one.
14-	section C.3: please expand on the algorithm (Chambolle & Pock, 2011) that can solve the inner min-max. Is it computationally efficient? Do you show anywhere in your simulations that tuning , ’ is going to reach global p*? I urge you to provide more details/intuitions about min-max objectives with coupled constraints. Do they arise naturally in min-max literature? If yes please provide some references.
15-	In the statements of your theorems you insist on using: “ for appropriate choice of regularizer”. I think the theorem should contain all the assumptions and choice of function/loss/regularizer. So please specify what type of regularizer you use in theorem statements.


**Summary Of The Paper:**

Min-max optimization problems for WGANs do not necessarily have saddle points. This paper aims at providing insights into optimization tractability of WGANs in the two-layer discriminator case with different activation functions.

**Summary Of The Review:**

This paper tries to shed light on the optimization challenges of the popular WGAN formulation, and the simplified assumptions on discriminator/generator architecture look like valid ones to me. Although the authors use the previously established results in other works, gathering and presenting such results seems to be very important for less understood areas such as non-convex non-concave GAN scenarios, and I enjoyed reading through all theoretical parts of the paper.

---

### Decision · Program_Chairs · 2022-01-20

**Decision:**

Accept (Poster)

**Comment:**

The authors provide a convexification for the GAN training via integral probability metrics induced by two-layer neural networks. The exposition relies on the convexification tools recently proposed by the Pilanci et al., and provides interesting insights to follow up in the future.